# A Unified Density Operator View of Flow Control and Merging

**Riccardo De Santi** [1 2]  **Malte Franke** [1]  **Ya-Ping Hsieh** [1]  **Andreas Krause** [1 2]

## Abstract

Recent progress in large-scale flow and diffusion models raised two fundamental algorithmic challenges: $(i)$ control-based reward adaptation of pre-trained flows, and $(ii)$ integration of multiple models, i.e., flow merging. While current approaches address them separately, we introduce a unifying probability-space framework that subsumes both as limit cases, and enables *reward-guided flow merging*, allowing principled, task-aware combination of multiple pre-trained flows (e.g., merging priors while maximizing drug-discovery utilities). Our formulation renders possible to express a rich family of *operators over generative models densities*, including intersection (e.g., to enforce safety), union (e.g., to compose diverse models), interpolation (e.g., for discovery), their reward-guided counterparts, as well as complex logical expressions via *generative circuits*. Next, we introduce **R**eward-Guided **F**low **M**erging (RFM), a mirror-descent scheme that reduces reward-guided flow merging to a sequence of standard fine-tuning problems. Then, we provide first-of-their-kind theoretical guarantees for reward-guided and *pure* flow merging via RFM. Ultimately, we showcase the capabilities of the proposed method on illustrative settings providing visually interpretable insights, and apply our method to high-dimensional de-novo molecular design and low-energy conformer generation.

## 1. Introduction

Large-scale generative modeling has recently progressed at an unprecedented pace, with flow (Lipman et al., 2022; 2024) and diffusion models (Sohl-Dickstein et al., 2015; Song & Ermon, 2019; Ho et al., 2020) delivering high-fidelity samples in chemistry (Hoogeboom et al., 2022),

biology (Corso et al., 2022), and robotics (Chi et al., 2024). However, adoption in real-world applications like scientific discovery led to two fundamental algorithmic challenges: $(i)$ reward-guided fine-tuning, i.e., adapting pre-trained flows to maximize downstream utilities (e.g., binding affinity) (e.g., Domingo-Enrich et al., 2024; Uehara et al., 2024b; De Santi et al., 2025a), and $(ii)$ flow merging (FM): integrating multiple pre-trained models into one (Song et al., 2023; Ma et al., 2025), e.g., to incorporate safety constraints (Dai et al., 2023), or unify diverse priors (Ma et al., 2025). Crucially, so far these two problems have been treated via fundamentally distinct formulations and methods. On the contrary, in this work we ask:

*Can adaptation and merging be cast in a single framework enabling task-aware merging of multiple flows?*

Intuitively, this would allow to merge flows in a task-aware manner, as well as adapt flows to optimize downstream rewards while leveraging information from multiple prior models. Answering this would contribute to the algorithmic-theoretical foundations of *flow adaptation*.

**Our approach** To address this challenge, we first introduce a probability-space optimization framework (see Fig. 1b) that recovers reward-guided fine-tuning and *pure* model merging as limit cases, and provably enables *reward-guided flow merging* (Sec. 3). Our formulation allows to express a rich family of operators over generative models, covering practical needs such as enforcing safety (e.g., via intersection), composing diverse models (e.g., via union), and discovery in data-scarce regions (e.g., via interpolation). However, these operators are expressed via non-linear functionals that cannot be optimized via classic RL or control schemes, as shown by De Santi et al. (2025a). To overcome this challenge, we introduce **R**eward-Guided **F**low **M**erging (RFM), a mirror descent (MD) (Nemirovskij & Yudin, 1983) scheme that solves reward-guided and pure flow merging via a sequential adaptation process implementable via established fine-tuning methods (e.g., Domingo-Enrich et al., 2024; Uehara et al., 2024b) (Sec. 4). Next, we extend the algorithm proposed, to operate on the space of entire flow processes, enabling scalable and stable computation of the intersection operator (Sec. 5). We provide a rigorous convergence analysis of RFM, yielding first-of-its-kind theoretical guarantees for reward-guided and pure flow merging (Sec. 6). Ultimately, we showcase our method's capabilities

---

[1]ETH Zurich [2]ETH AI Center. Correspondence to: Riccardo De Santi <rdesanti@ethz.ch>.

*Proceedings of the 43rd International Conference on Machine Learning*, Seoul, South Korea. PMLR 306, 2026. Copyright 2026 by the author(s).

on illustrative settings, as well as on high-dimensional molecular design and conformer generation tasks (Sec. 7).

**Our contributions**  To sum up, in this work we contribute

- A formalization of *reward-guided flow merging* via density operators, which unifies and generalizes reward-guided adaptation and flow merging (Sec. 3).
- *Reward-Guided Flow Merging (*RFM*)*, a probability-space optimization algorithm that provably solves reward-guided flow merging, and pure flow merging problems (Sec. 4), and a stability-enhancing extension for flow intersection, implementing a scalable mirror-descent scheme over the space of flow processes (Sec. 5).
- Theoretical convergence guarantees for the proposed algorithms, leveraging recent insights on mirror flows, and a novel technical result of independent interest. (Sec. 6).
- An experimental evaluation of RFM showcasing its practical relevance on both synthetic, yet illustrative settings, as well as on scientific discovery tasks, namely molecular design and conformer generation. (Sec. 7).

## 2. Background and Notation

We denote the set of Borel probability measures on a set $\mathcal{X}$ with $P(\mathcal{X})$, and the set of functionals over $P(\mathcal{X})$ as $F(\mathcal{X})$.

**Generative Flow Models.**  Generative models aim to approximately sample novel data points from a data distribution $p_{data}$. Flow models tackle this problem by transforming samples $X_0 = x_0$ from a source distribution $p_0$ into samples $X_1 = x_1$ from the target distribution $p_{data}$ (Lipman et al., 2024; Farebrother et al., 2025). Formally, a *flow* is a time-dependent map $\psi : [0,1] \times \mathbb{R}^d \to \mathbb{R}$ such that $\psi : (t, x) \to \psi_t(x)$. A *generative flow model* is a continuous-time Markov process $\{X_t\}_{0 \le t \le 1}$ obtained by applying a flow $\psi_t$ to $X_0 \sim p_0$ as $X_t = \psi_t(X_0)$, $t \in [0,1]$, such that $X_1 = \psi_1(X_0) \sim p_{data}$. In particular, the flow $\psi$ can be defined by a *velocity field* $u : [0,1] \times \mathbb{R}^d \to \mathbb{R}^d$, which is a vector field related to $\psi$ via the following ordinary differential equation (ODE), typically referred to as *flow ODE*:

$$\frac{\mathrm{d}}{\mathrm{d}t}\psi_t(x) = u_t(\psi_t(x)) \qquad (1)$$

with initial condition $\psi_0(x) = 0$. A flow model $X_t = \psi_t(X_0)$ induces a probability path of *marginal densities* $p = \{p_t\}_{0 \le t \le 1}$ such that at time $t$ we have that $X_t \sim p_t$. We denote by $p^u$ the probability path of marginal densities induced by the velocity field $u$. Flow matching (FM) (Lipman et al., 2024) can estimate a velocity field $u^\theta$ s.t. the induced marginal densities $p^{u_\theta}$ satisfy $p_0^{u_\theta} = p_0$ and $p_1^{u_\theta} = p_{data}$, where $p_0$ denotes the source distribution, and $p_{data}$ the target data distribution. Typically FM are rendered tractable by defining $p_t^u$ as the marginal of a conditional density $p_t^u(\cdot|x_0, x_1)$, e.g.,:

$$X_t \mid X_0, X_1 = \kappa_t X_0 + \omega_t X_1 \qquad (2)$$

where $\kappa_0 = \omega_1 = 1$ and $\kappa_1 = \omega_0 = 0$ (e.g. $\kappa_t = 1 - t$ and $\omega_t = t$). Then $u^\theta$ can be learned by regressing onto the conditional velocity field $u(\cdot|x_1)$ (Lipman et al., 2022). As diffusion models (Song & Ermon, 2019) (DMs) admit an equivalent ODE formulation (Lipman et al., 2024, Ch. 10), our contributions extend directly to DMs.

**Continuous-time Reinforcement Learning.**  We formulate finite-horizon continuous-time RL as a specific class of optimal control problems (Wang et al., 2020; Zhao et al., 2024). Given a state space $\mathcal{X}$ and an action space $\mathcal{A}$, we consider the transition dynamics governed by the following ODE:

$$\frac{\mathrm{d}}{\mathrm{d}t}\psi_t(x) = a_t(\psi_t(x)) \qquad (3)$$

where $a_t \in \mathcal{A}$ is an action. We consider a state space $\mathcal{X} := \mathbb{R}^d \times [0,1]$, and denote by (Markovian) deterministic policy a function $\pi_t(X_t) := \pi(X_t, t) \in \mathcal{A}$ mapping a state $(x,t) \in \mathcal{X}$ to an action $a \in \mathcal{A}$ such that $a_t = \pi(X_t, t)$, and denote with $p_t^\pi$ the marginal density at time $t$ induced by policy $\pi$.

**Pre-trained Flow Models as an RL policy.**  A pre-trained flow model with velocity field $u^{pre}$ can be interpreted as an action process $a_t^{pre} := u^{pre}(X_t, t)$, where $a_t^{pre}$ is determined by a continuous-time RL policy via $a_t^{pre} = \pi^{pre}(X_t, t)$ (De Santi et al., 2025b). Therefore, we can express the flow ODE induced by a pre-trained flow model by replacing $a_t$ with $a^{pre}$ in Eq. equation 3, and denote the pre-trained model by its policy $\pi^{pre}$, which induces a density $p_1^{pre} := p_1^{\pi^{pre}}$ approximating $p_{data}$.

## 3. Reward-Guided FM via Density Operators

In this section, we introduce the general problem of *reward-guided flow merging* via *density operators*. Formally, we wish to implement an operator $\mathcal{O} : \Pi \times \ldots \times \Pi \to \Pi$ that, given pre-trained generative flow models $\{\pi^{pre,i}\}_{i \in [n]}$, returns a merged flow $\pi^*$ inducing an ODE:

$$\frac{\mathrm{d}}{\mathrm{d}t}\psi_t(x) = a_t^*(\psi_t(x)) \quad \text{with} \quad a_t^* = \pi^*(x, t), \qquad (4)$$

such that it controllably merges prior information within the $n$ pre-trained generative models, while potentially steering its density $p_1^* := p_1^{\pi^*}$ towards a high-reward region according to a given scalar reward function $f(x) : \mathcal{X} \to \mathbb{R}$. We implement such operators by fine-tuning an initial flow $\pi^{init} \in \{\pi^{pre,i}\}_{i \in [n]}$ according to the following probability-space optimization problem, visually portrayed in Fig. 1b.

**Reward-Guided FM via Density Operator**

$$\pi^* \in \underset{\pi : p_0^* = p_0^{pre}}{\arg\max} \; \underset{x \sim p_1^\pi}{\mathbb{E}} \left[ f(x) \right] - \sum_{i=1}^n \alpha_i \mathcal{D}_i(p_1^\pi \, \| \, p_1^{pre,i})$$

$$(5)$$

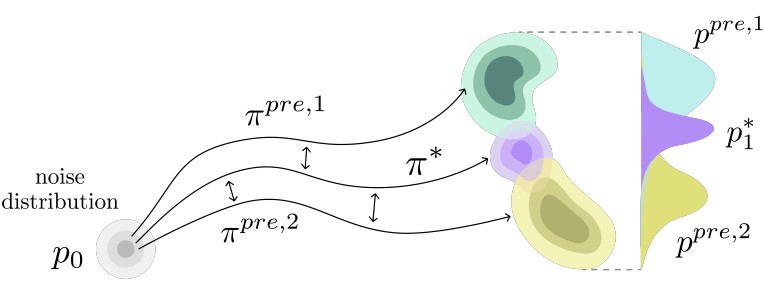
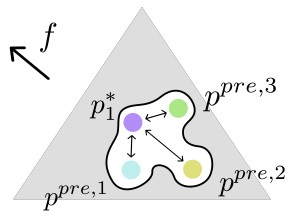

*(a)* Reward-Guided Flow Merging        *(b)* Probability-Space Opt. Viewpoint

*Figure 1.* (1a) Pre-trained and fine-tuned policies inducing $\{p_1^{pre,i}\}_{i=1}^n$ and optimal density $p_1^*$ computed via flow merging, i.e., subcase of Problem 5 where $f$ is disregarded. (1b) Probability-space optimization viewpoint on reward-guided flow merging, as in Problem 5.

Here, each $D_i$ is an arbitrary divergence, $\alpha_i > 0$ are model-specific weights, and $p_0^\pi = p_0^{pre}$ enforces that the marginal density at $t = 0$ must match the pre-trained model marginal. This formulation recovers reward-guided fine-tuning (e.g., Domingo-Enrich et al., 2024) when $n = 1$ and $\mathcal{D}_1 = D_{KL}$, and provides a formal framework for *pure* flow merging (e.g., Poole et al., 2022; Song et al., 2023) with interpretable objectives, when the reward $f$ is constant (e.g., $f(x) = 0 \ \forall x \in \mathcal{X}$). In this case, Eq. 5 formalizes flow merging as computing a flow $\pi^*$ that minimizes a weighted sum of divergences to the priors $\{\pi^{pre,i}\}_{i \in [n]}$. Varying the divergences $\{D_i\}_{i \in [n]}$ yields different merging strategies.

**In-Distribution Flow Merging.** Given pre-trained flow models $\{\pi^{pre,i}\}_{i \in [n]}$, we denote by *in-distribution* merging when the merged model generates samples from regions with sufficient prior density. Practically relevant instances include the *intersection operator* $\mathcal{O}_\wedge$ (i.e., a logical AND), and the *union operator* $\mathcal{O}_\vee$ (i.e., a logical OR). Formally, these operators can be defined via the following expressions:

$\mathcal{O}_\wedge$: **Intersection ($\wedge$) Operator**

$$\pi^* \in \underset{\pi:p_0^*=p_0^{pre}}{\arg\min} \sum_{i=1}^n \alpha_i \, D_{KL}(p_1^\pi \| p_1^{pre,i}) \quad (6)$$

$\mathcal{O}_\vee$: **Union ($\vee$) Operator**

$$\pi^* \in \underset{\pi:p_0^*=p_0^{pre}}{\arg\min} \sum_{i=1}^n \alpha_i \, D_{KL}^R(p_1^\pi \| p_1^{pre,i}) \quad (7)$$

The $D_{KL}$ divergences in Eq. 6 heavily penalize density allocation in any region with low prior density for any model $\pi^{pre,i}$, leading to an optimal flow model $\pi^*$ inducing $p_1^*(x) \propto \prod_{i=1}^n p_1^{pre,i}(x)^{\alpha_i}$ (cf. Heskes, 1997). Similarly, the reverse KL divergence $D_{KL}^R(p\|q) := D_{KL}(q\|p)$ in Eq. 7 induces a mode-covering behaviour implying a flow model $\pi^*$ with density $p_1^* \propto \sum_{i=1}^n \alpha_i p_1^{pre,i}(x)$ (cf. Banerjee et al., 2005) sufficiently covering all regions with enough prior density, for any $p_1^{pre,i}, \ i \in [n]$.

**Out-of-Distribution Flow Merging.** We denote by *out-of-distribution*, the case where $\pi^*$ samples from regions insufficiently covered by all priors. An example is the *interpolation operator* $\mathcal{O}_{W_p}$ (see Eq. 8), inducing $p_1^*$ equal to the priors Wasserstein Barycenter (Cuturi & Doucet, 2014).

$\mathcal{O}_{W_p}$: **Interpolation (Wasserstein-$p$ Barycenter) Operator**

$$\underset{\pi}{\arg\min} \sum_{i=1}^n \alpha_i W_p(p_1^\pi \| p_1^{pre,i}) \quad (8)$$

**Straightforward Generalizations.** While we presented a few practically relevant operators, the framework in Eqs. 5 is not tied to them: it trivially admits any new operator defined via other divergences (e.g., MMD, Rényi, Jensen–Shannon), and allows diverse $D_i$ for each prior flow models $\pi^{pre,i}$. Moreover, sequential composition of these operators makes it possible to implement arbitrarily complex logical operations over generative models. For instance, as later shown in Sec. 7, one can obtain $\pi^* = (\pi^{pre,1} \vee \pi^{pre,2}) \wedge \pi^{pre,3}$ by first computing $\pi_{1,2} := \mathcal{O}_\vee(\pi^{pre,1}, \pi^{pre,2})$ and then $\pi^* := \mathcal{O}_\wedge(\pi_{1,2}, \pi^{pre,3})$. We denote such operators by *generative circuits*, and illustrate one in Fig. 3d.

While being of high practical relevance, the presented framework entails optimizing non-linear distributional utilities (see Eq. 5) beyond the reach of standard RL or control schemes, as shown by De Santi et al. (2025a). In the next section, we show how to reduce the introduced problem to sequential fine-tuning for maximization of rewards automatically determined by the choice of operator $\mathcal{O}$.

## 4. Algorithm: Reward-Guided Flow Merging

In this section, we introduce **R**eward-**G**uided **F**low **M**erging (RFM), see Alg. 1, which provably solves Problem 5 by solving the probability-space optimization problem:

$$p_1^{\pi^*} \in \underset{p_1^\pi}{\arg\max} \, \mathcal{G}(p_1^\pi) \quad (9)$$

**Algorithm 1 R**eward-Guided **F**low **M**erging (RFM)

---

1: **input:** $\{\pi^{pre,i}\}_{i\in[n]}$ : pre-trained flows, $\{\mathcal{D}_i\}_{i\in[n]}$ : arbitrary divergences, $f$ : reward, $\{\alpha_i\}_{i\in[n]}$ : weighs, $K$ : iterations number, $\{\gamma_k\}_{k=1}^K$ stepsizes, $\pi^{init} \in \{\pi^{pre,i}\}_{i\in[n]}$ : initial flow model
2: **Init:** $\pi_0 := \pi^{init}$, $p_1^k := p_1^{\pi^k}$
3: **for** $k = 1, 2, \ldots, K$ **do**
4:     Estimate $\nabla_x g_k = \nabla_x \delta\mathcal{G}(p_1^{k-1})$ with:

$$\mathcal{G}\left(p_1^{k-1}\right) = \mathop{\mathbb{E}}_{x \sim p_1^{k-1}}[f(x)] - \sum_{i=1}^n \alpha_i \mathcal{D}_i(p_1^{k-1} \| p_1^{pre,i})$$

5:     Compute $\pi_k$ via standard reward-guided fine-tuning:

$$\pi_k \leftarrow \textsc{RewardGuidedFTSolver}(\nabla_x g_k, \gamma_k, \pi_{k-1})$$

6: **end for**
7: **output:** policy $\pi := \pi_K$

---

where the functional $\mathcal{G}(p_1^\pi)$ is defined as:

$$\mathcal{G}(p_1^\pi) := \mathop{\mathbb{E}}_{x \sim p_1^\pi}[f(x)] - \sum_{i=1}^n \alpha_i \mathcal{D}_i(p_1^\pi \| p_1^{pre,i}) \qquad (10)$$

Given an initial flow model $\pi^{init} \in \{\pi^{pre,i}\}_{i\in[n]}$, RFM follows a mirror descent (MD) scheme (Nemirovskij & Yudin, 1983) for $K$ iterations by sequentially fine-tuning $\pi^{init}$ to maximize surrogate rewards $g_k$ determined by the chosen operator, i.e., $\mathcal{G}$. To understand how RFM computes the surrogate rewards $\{g_k\}_{k=1}^K$ guiding the optimization process in Eq. 10, we first recall the notion of first variation of $\mathcal{G}$ over a space of probability measures (cf. Hsieh et al., 2019). A functional $\mathcal{G} \in \mathbf{F}(\mathcal{X})$ has a first variation at $\mu \in \mathbf{P}(\mathcal{X})$ if there exists a function $\delta\mathcal{G}(\mu) \in \mathbf{F}(\mathcal{X})$ such that:

$$\mathcal{G}(\mu + \epsilon\mu') = \mathcal{G}(\mu) + \epsilon\langle\mu', \delta\mathcal{G}(\mu)\rangle + o(\epsilon).$$

holds for all $\mu' \in \mathbf{P}(\mathcal{X})$, where the inner product is an expectation. At iteration $k \in [K]$, given the current generative model $\pi^{k-1}$, RFM fine-tunes it according to the following standard entropy-regularized control or RL problem, solvable via any established method (e.g., Domingo-Enrich et al., 2024)

$$\mathop{\arg\max}_\pi \quad \langle\delta\mathcal{G}\left(p_1^{\pi_{k-1}}\right), p_1^\pi\rangle - \frac{1}{\gamma_k}D_{KL}(p_1^\pi \| p_1^{\pi_{k-1}}) \quad (11)$$

Thus, we introduce a surrogate reward function $g_k : \mathcal{X} \to \mathbb{R}$ defined for all $x \in \mathcal{X}$ such that:

$$g_k(x) := \delta\mathcal{G}\left(p_1^{k-1}\right)(x) \qquad (12)$$

$$\mathop{\mathbb{E}}_{x \sim p_1^\pi}[g_k(x)] = \langle\delta\mathcal{G}\left(p_1^{k-1}\right), p_1^\pi\rangle \qquad (13)$$

We now present **R**eward-Guided **F**low **M**erging (RFM), see Alg. 1. At each iteration $k \in [K]$, RFM estimates the gradient of the first variation at the previous policy $\pi_{k-1}$, i.e., $\nabla_x \delta\mathcal{G}(p_1^{\pi_{k-1}})$ (line 4). Then, it updates the flow model $\pi_k$ by solving the reward-guided fine-tuning problem in

Eq. 11 by employing $\nabla_x g_k := \nabla_x \delta\mathcal{G}(p_1^{\pi_{k-1}})$ as reward function gradient (line 5). Ultimately, RFM returns a final policy $\pi := \pi_K$. We report a detailed implementation of REWARDGUIDEDFTSOLVER in Apx. E.2.

**Implementation of Intersection, Union, and Interpolation operators.** We present the specific expressions of $\nabla_x \delta\mathcal{G}(p_1^\pi)$ for pure model merging respectively for the intersection ($\mathcal{O}_\wedge$), union ($\mathcal{O}_\vee$), and interpolation ($\mathcal{O}_{W_p}$) operators introduced in Sec. 3, namely, $\nabla_x \delta\mathcal{G}(p_1^\pi)(x) =$

$$\begin{cases} -\sum_{i=1}^n \alpha_i s^{k-1}(x, t=1) + \sum_{i=1}^n \alpha_i s^{\pi^{pre,i}}(x, t=1) \\ -\sum_{i=1}^n \nabla_x \exp\left(\phi_i^*(x) - 1\right), \phi_i^* \text{ as by Eq. } 46 \\ -\sum_{i=1}^n \nabla_x \phi_i^*(x), \phi_i^* = \arg\max_{\phi: \|\nabla_x\phi\|\le 1}\langle\phi, p^\pi - p^{pre,i}\rangle \end{cases}$$

Where by $s^{k-1}(x,t) := \nabla \log p_t^{\pi-1}(x)$ we denote the score of model $\pi^{k-1}$ at point $x$ and time $t$, and $s^{pre,i} := s^{\pi^{pre,i}}$. For diffusion models, a learned neural score network is typically available; for flows, the score follows a standard linear transformation of $\pi(X_t, t)$ (e.g., Domingo-Enrich et al., 2024, Eq. 8):

$$s_t^\pi(x) = \frac{1}{\kappa_t(\frac{\dot\omega_t}{\omega_t}\kappa_t - \dot\kappa_t)}\left(\pi(x, t) - \frac{\dot\omega_t}{\omega_t}x\right) \qquad (14)$$

In Apx. C.2, we report the gradient expressions above, and present a brief tutorial to derive the first variations for any new operator. Moreover, in Apx. D, we introduce a scalable method to reduce learning of multiple critics $\phi_i^*$ to learning only one for the case of union operator with multiple priors, and prove its correctness in Prop. 1.

Crucially, the score in Eq. 14 for the intersection gradient, diverges at $t = 1$. In the following, we propose a principled solution to this problem by leveraging weighted score estimates along the entire noised flow process, i.e., $t \in [0, 1)$.

## 5. Scalable Intersection via Flow Optimization

To tackle the aforementioned issue, we lift the problem in Eq. 6 from the probability space of the last time-step marginal $p_1^\pi$, where the score diverges, to the entire flow process:

> **Intersection Operator via Flow Process Optimization**
>
> $$\mathop{\arg\max}_{\pi: p_0^\pi = p_0^{pre}} \mathcal{L}_\wedge(\mathbf{Q}^\pi) := \int_0^1 \lambda_t \sum_{i=1}^n \alpha_i D_{KL}(p_t^\pi \| p_t^{pre,i})\, \mathrm{d}t$$
>
> $$(15)$$

Here, $\mathbf{Q}^\pi = \{p_t^\pi\}_{t\in[0,1]}$ denotes the entire joint flow process induced by policy $\pi$ over $\mathcal{X}^{[0,1]}$. Under general regularity assumptions, an optimal policy $\pi^*$ for Problem 15 is optimal also w.r.t. Eq. 6. Interestingly, an optimal flow $\pi^*$ for Problem 15 can be computed via a MD scheme acting over the space of joint flow processes $\mathbf{Q}^\pi = \{p_t^\pi\}_{t\in[0,1]}$ determined by the following update rule:

**Reward-Guided FM (Mirror Descent) Step**

$$\mathbf{Q}^k \in \underset{q:p_0=p_0^{k-1}}{\arg\max} \langle \delta\mathcal{L}_\wedge(\mathbf{Q}^{k-1}), \mathbf{Q}\rangle + \frac{1}{\gamma^k} D_{KL}\left(\mathbf{Q}\|\mathbf{Q}^{k-1}\right) \tag{16}$$

First, we state the following Lemma 5.1, which allows to express the first variation of $\mathcal{L}_\wedge$ w.r.t. the flow process $\mathbf{Q}^\pi$ as an integral of first variations w.r.t. marginal densities $p_t^\pi$.

**Lemma 5.1** (First Variation of Flow Process Functional). *For objective $\mathcal{L}_\wedge$ in Eq. 15 it holds the following:*

$$\langle \delta\mathcal{L}_\wedge(\mathbf{Q}^k), q\rangle = \int_0^1 \lambda_t \; \mathbb{E}_\mathbf{Q}\left[\delta \sum_{i=1}^n \alpha_i \, D_{KL}(p_t^\pi \| p_t^{pre,i})\right] \mathrm{d}t \tag{17}$$

This factorization of $\langle \delta\mathcal{L}_\wedge(\mathbf{Q}^k), q\rangle$ shows that a flow $\pi_{k+1}$ inducing an optimal process $\mathbf{Q}^k$ w.r.t. the update step in Eq. 16 can be computed by solving a control-affine optimal control problem via the same REWARDGUIDEDFTSOLVER oracle used in Alg. 1, by introducing the running cost term:

$$f_t(x) := \delta\left(\sum_{i=1}^n \alpha_i \, D_{KL}(p_t^\pi \| p_t^{pre,i})\right)(x,t) \tag{18}$$

with $\in [0,1)$. This algorithmic idea, which allows to control the score scale at $t \to 1$ via $\lambda_t$, thus enhancing RFM, trivially extends to reward-guided merging, and is accompanied by a detailed pseudocode in Apx. E.2.

# 6. Guarantees for Reward-Guided FM

In this section, we aim to provide rigorous convergence guarantees for RFM by interpreting its iterations as *mirror descent* on the space of measures. To this end, at each iteration it must hold that $s^{k-1}(x,t) := \nabla_x \log p_t^{\pi^{k-1}}(x)$, i.e., the REWARDGUIDEDFTSOLVER subroutine, employed by RFM at every iteration (see line 5, Alg. 1), *retains the score information*. Our key contribution is to prove this result, which is not only essential for our convergence analysis, but also of independent interest: it provides a rigorous justification for a structural assumption that underlies several control-based fine-tuning analyses, where it is typically implicitly assumed (De Santi et al., 2025a;b). Thus, the convergence result below should be read as a conditional inexact-oracle guarantee: it applies when the first variation defining the surrogate reward can be evaluated or consistently approximated, and when the fine-tuning solver satisfies the noise and bias conditions stated in Apx. B.2.

**Score Retention via Control-based Fine-Tuning.** Our main observation is that, in the continuous-time stochastic optimal control (SOC) idealization, control-based fine-tuning schemes retain score information. This includes the idealized counterpart of Adjoint Matching (Domingo-Enrich et al., 2024), which we utilize in our

experiments (Sec. 7). We consider the standard SOC framework, exposed in Apx. B, and show that the exact SOC solution encodes the score information required by the mirror-descent interpretation. In finite-step, discretized neural implementations, departures from exact terminal noising should be interpreted as additional approximation error in the inexact-update analysis below.

**Theorem 6.1** (SOC Retains Score Information). *Suppose that the prior diffusion model forward process converges to a standard Gaussian noise[a]. Then, the model returns by a SOC fine-tuning solver is such that:*

$$u^\star(x,t) := \sigma(t)\, \nabla \log p_t^k(x) \tag{19}$$

*where $p_t^k$ denotes the marginal distribution of the prior forward process, initialized at $p_1^{\pi^k}$, and $u^\star(x,t)$ the applied optimal control (see Sec. B). In other words,* REWARDGUIDEDFTSOLVER *exactly recovers the score.*

---
[a]This is a standard assumption in diffusion modeling

Theorem 6.1 enables us to reinterpret Algorithm 1 as generating *approximate mirror iterates*, a framework that has proven effective for sampling and generative modeling (Karimi et al., 2024; De Santi et al., 2025b;a).

**Robust Convergence under Inexact Updates.** Thanks to Theorem 6.1, we can now develop a rigorous convergence theory for Algorithm 1 under the realistic condition that REWARDGUIDEDFTSOLVER (see Sec. 4) is implemented *approximately*. Let $\mathcal{G}$ be the objective in Eq. 10. Via $\pi^k$, the iterates generated by Algorithm 1 induce a sequence of stochastic processes, denoted by $\mathbf{Q}^k$, which satisfy $\mathbf{Q}^k = p_1^{\pi^k}$. Each iterate $\mathbf{Q}^k$ is understood as an approximation to the *idealized* mirror descent step:

$$\mathbf{Q}_\sharp^k \in \underset{\mathbf{Q}:p_0=p_0^{pre}}{\arg\max}\left\{\langle \delta\mathcal{G}(p_1^{\pi^k}), \mathbf{Q}\rangle - \frac{1}{\gamma^k}D_{KL}\left(\mathbf{Q} \| \mathbf{Q}^{k-1}\right)\right\}. \tag{20}$$

which serves as the exact reference point for our analysis. To quantify the discrepancy between $\mathbf{Q}^k$ and $\mathbf{Q}_\sharp^k$, let $\mathcal{T}_k$ denote the history up to step $k$, and decompose the error as

$$b_k := \mathbb{E}\left[\delta\mathcal{G}(p_1^{\pi_k}) - \delta\mathcal{G}((\mathbf{Q}_\sharp^k)_1)\,\big|\,\mathcal{T}_k\right], \tag{21}$$

$$U_k := \delta\mathcal{G}(p_1^{\pi_k}) - \delta\mathcal{G}((\mathbf{Q}_\sharp^k)_1) - b_k. \tag{22}$$

Here, $b_k$ captures systematic approximation error, and $U_k$ represents a zero-mean fluctuation conditional on $\mathcal{T}_k$. Under precompactness, noise, and bias assumptions stated in Section B.2, which model the approximation error introduced by using an approximate reward-guided fine-tuning solver, the long-term behavior of the iterates can be characterized.

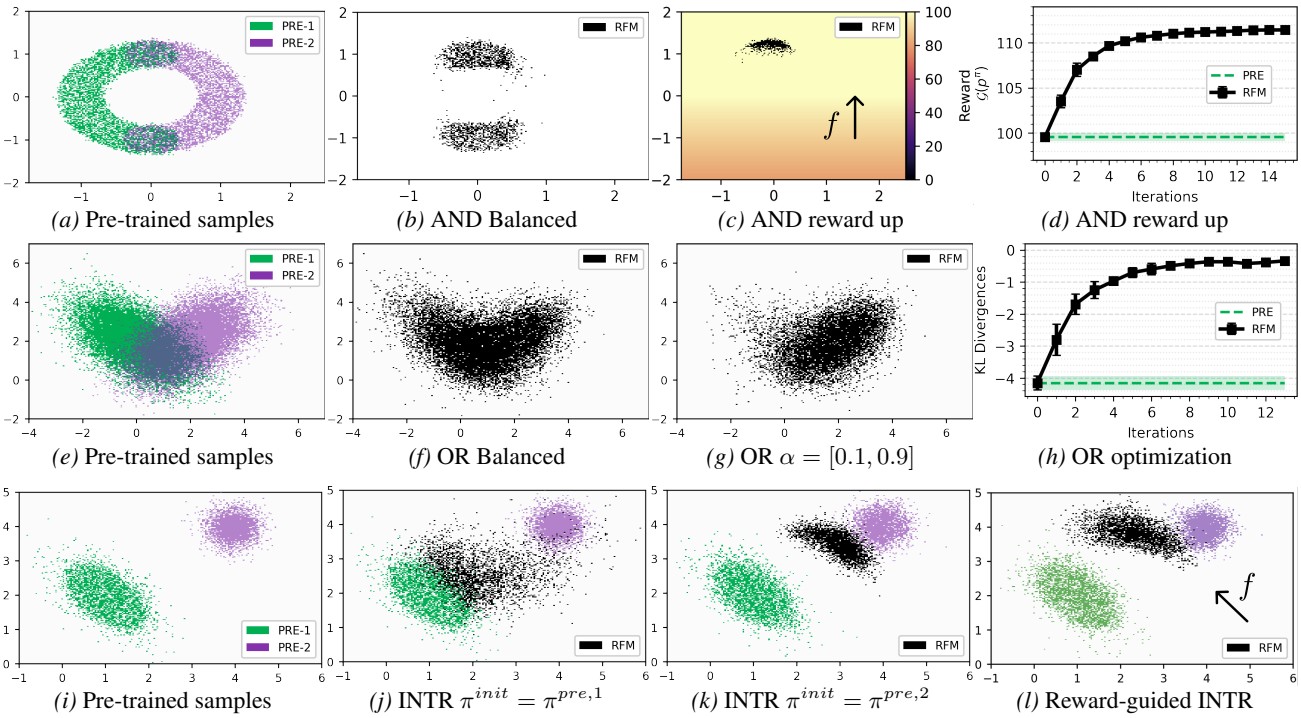

*Figure 2.* Illustrative settings with visually interpretable results. (top) Flow model balanced pure intersection (2b), and reward-guided intersection (2c), (mid) Flow balanced and unbalanced union, (bottom) Flow model pure and reward-guided interpolation. Crucially, RFM can correctly implement these practically relevant and diverse operators with high degree of expressivity (e.g., $\alpha$, reward-guidance).

**Theorem 6.2** (Asymptotic convergence under inexact updates (Informal))**.** *Assume the oracle has bounded variance and diminishing bias, and the step sizes $\{\gamma^k\}$ satisfy the Robbins–Monro conditions ($\sum_k \gamma^k = \infty$, $\sum_k (\gamma^k)^2 < \infty$). Then the sequence $\{p_1^{\pi_k}\}$ generated by Algorithm 1 converges almost surely to the optimum in the weak sense:*

$$p_1^{\pi_k} \rightharpoonup \tilde{p}_1 \quad a.s., \tag{23}$$

*where $\tilde{p}_1$ is a stationary point of $\mathcal{G}$.*

**Remark.** Several functionals $\mathcal{G}$ considered in this work (e.g., forward and reverse KL, Jensen–Shannon, and their $f$-guided counterparts etc.) are convex over the space of measures. In these cases, Thm. 6.2 trivially strengthens to convergence to a *global* optimum: $p_1^{\pi_k} \rightharpoonup p_1^\star = \mathbf{Q}_1^\star$ for some $\mathbf{Q}^\star \in \arg\max_{\mathbf{Q}:\, \mathbf{Q}_0 = p_0^{\mathrm{pre}}} \mathcal{G}(\mathbf{Q}_1)$, as shown in Apx. B.2.

## 7. Experimental Evaluation

We evaluate RFM for the reward-guided flow merging problem (see Eq. 5) by tackling two types of experiments: (i) visually interpretable illustrative settings, showcasing the correctness and high expressivity of RFM, and (2) high-dimensional molecular design and conformer generation tasks. Further experimental details are reported in Apx. I

**Intersection Operator $\mathcal{O}_\wedge$.** We consider pre-trained flow models inducing densities $p_1^{pre,1}$ (green) and $p_1^{pre,2}$ (violet),

as shown in Fig. 2a. We fine-tune $\pi^{init} := \pi^{pre,1}$ via RFM to compute the policy $\pi^*$ resulting from diverse intersection operations $\pi^* = \mathcal{O}_\wedge(\pi^{pre,1}, \pi^{pre,2})$. First, in Fig. 2b, we show $p^*$ (black) obtained by RFM with $\alpha = [0.5, 0.5]$, i.e., *balanced* (B). One can notice that the flow model $p^*$ covers mostly the intersecting regions between $p_1^{pre,1}$ and $p_1^{pre,2}$ (see Fig. 2a). In Fig. 2c we report an instance of reward-guided intersection (RG) for a reward function maximized upward. As one can see, RFM computes a policy $\pi^*$ placing density over the highest-reward region among the intersecting ones, i.e., the top intersecting area. This reward-guided flow merging process is carried out via maximization over $K = 15$ iterations of the objective $\mathcal{G}$ illustrated in Fig. 2d.

**Union Operator $\mathcal{O}_\vee$.** We fine-tune the pre-trained flow model $\pi^{init} = \pi^{pre,1}$ with density illustrated in Fig. 2e (green) via RFM to implement balanced (i.e., $\alpha = [0.5, 0.5]$ and unbalanced (i.e., $\alpha = [0.1, 0.9]$ (UB)) versions of the union operator, namely computing $\pi^* = \mathcal{O}_\vee(\pi^{pre,1}, \pi^{pre,2})$. As shown in Fig. 2f and 2g RFM can successfully compute optimal policies $\pi^*$ implementing both operators via optimization of the functional $\mathcal{G}$, corresponding to sum of weighted KL-divergences (see Eq. 7) evaluated for iterations $k \in [K]$ with $K = 13$ in Fig. 2h.

**Interpolation Operator $\mathcal{O}_{W_1}$.** We use RFM to compute flows $\pi^*$ inducing $p_1^*$ corresponding to diverse interpolations between the the pre-trained models' densities illustrated in Fig. 2i. Although the optimal policy to which RFM con-

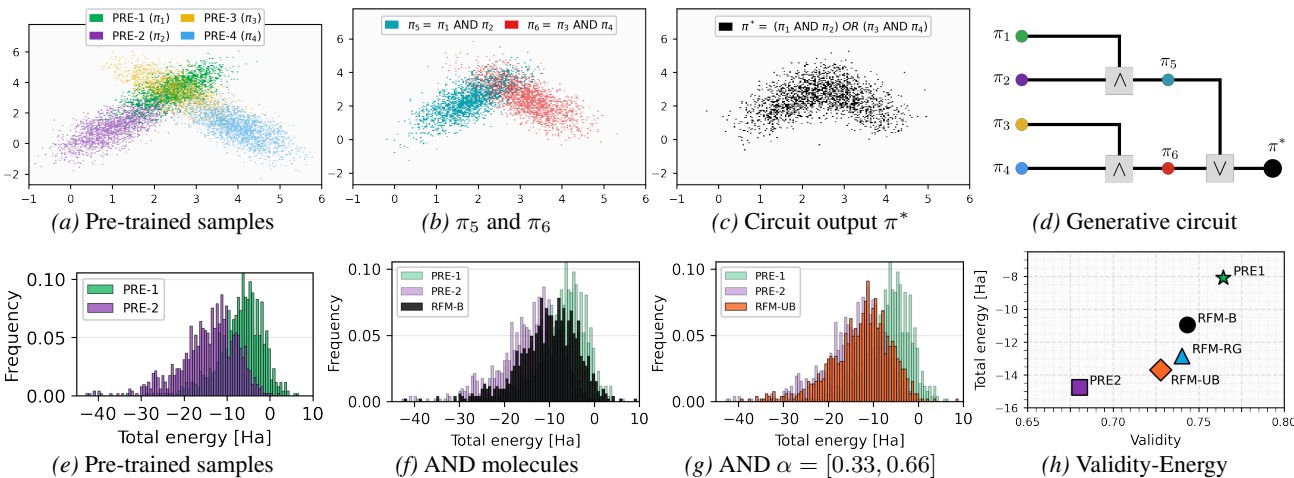

*Figure 3.* (top) RFM implements a generative circuit (3d) describing a complex logical expressions ($\pi^* = (\pi_1 \wedge \pi_2) \vee (\pi_3 \wedge \pi_4)$) by computing sequential operators (3a-3c). (bottom) RFM computes a flows intersection $\pi^*$ generating drug molecules with desired energy levels.

verges asymptotically is invariant w.r.t. the initial flow model $\pi^{init}$ chosen for fine-tuning, here we show that this choice can actually be used to control the algorithm execution over few iterations (i.e., $K = 6$). As one can expect, Fig. 2j and 2k show that the result density after $K = 6$ iterations is closer to the flow model chosen as $\pi^{init}$, namely $\pi^{pre,1}$ (green) in Fig. 2j and $\pi^{pre,2}$ (violet) in Fig. 2k. We illustrate in Fig. 2l the density (black) obtained via reward-guided interpolation, with a reward function maximized left upwards.

**Complex Logical Expressions via Generative Circuits.**
We consider 4 flow models $\{\pi_{pre,i}\}_{i=1}^4$ as in Fig. 3a, which we aim to merge into one flow $\pi^*$ determined by the logical expression $\pi^* = (\pi_1 \wedge \pi_2) \vee (\pi_3 \wedge \pi_4)$. We implement the generative circuit in Fig. 3d via sequential use of RFM. First, we compute $\pi_5 := \mathcal{O}_\wedge(\pi^{pre,1}, \pi^{pre,2})$ and $\pi_6 := \mathcal{O}_\wedge(\pi^{pre,3}, \pi^{pre,4})$, shown in Fig. 3b, and subsequently $\pi^* := \mathcal{O}_\vee(\pi^{pre,3}, \pi^{pre,4})$, as illustrated in Fig. 3c. This illustrative experiments confirms that RFM can implement complex logical expressions over generative models via generative circuits, as the simple one just presented.

**Low-Energy Molecular Design via Flow Merging**
Navigating chemical space to discover novel structures with desirable properties is a central goal of data-driven molecular design. A generative model must produce diverse, chemically valid structures that follow specified property profiles and constraints. We base our case study on two FlowMol models $\pi^{pre,1}$ and $\pi^{pre,2}$ (Dunn & Koes, 2024) pre-trained on GEOM-Drugs (Axelrod & Gomez-Bombarelli, 2022) with different levels of single-point total energy at GFN1-xTB level of theory (Friede et al., 2024), $-14.8$ and $-8.1$ Ha as shown in Fig. 3e. We aim to compute a flow model that generates molecules whose total energy is likely under both generative models. To this end, we run RFM to compute the flow $\pi^*$ returned by the intersection operator (see Eq. 6), with parameters detailed in Apx. I. We report the density

$p_1^*$ (black) computed via balanced merging (i.e., $\alpha_i = 1$) in Fig. 3f and the one obtained via unbalanced merging (i.e., $\alpha_1 = 1, \alpha_2 = 2$) in Fig. 3g. In the former case, $p_1^*$ correctly places the majority of its density on the overlapping region between the two priors within $[-20, 0]$ Ha (see Fig. 3f). The estimated mean energy of $\pi^*$ (black) i.e., $-10.95 \pm 0.28$ Ha, reported along with validity in 3h matches the energy value of maximal overlap between $\pi^{pre,1}$ and $\pi^{pre,2}$, as one can see in 3e. Adding reward-guidance leads to lower energy values compared to the balanced merging model while keeping its high validity. In the unbalanced case, RFM shifts the density slightly leftwards, effectively implementing the $\alpha$-weighted intersection. We report energy-validity metrics resulting from balanced and unbalanced intersection in Fig. 3h, and compare them with their reward-guided counterpart in Table 1. Next, we compute via RFM the union operator over two FlowMol pre-trained on the QM9 dataset (Ramakrishnan et al., 2014). We parametrize critics $\phi_i^*$ (see Sec. 1) via the FlowMol latent representation with an MLP readout layer. Figure 7 shows that the estimated mean of the model $\pi^*$ obtained via RFM matches the average total energy of $\pi^{pre,1}$ and $\pi^{pre,2}$ as predicted by the closed-form expression for union from Sec. 3.

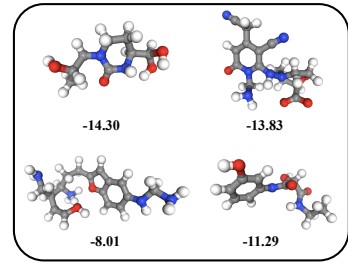

*Figure 4.* Drug-like molecules generated by $\pi^*_{AND}$ flow via RFM.

**Reward-Guided FM of Conformer Generation Models**
Inferring 3D conformers from a molecule's topology is a key prerequisite for many computational chemistry applications including molecular docking (McNutt et al., 2023), thermodynamic property prediction (Pracht &

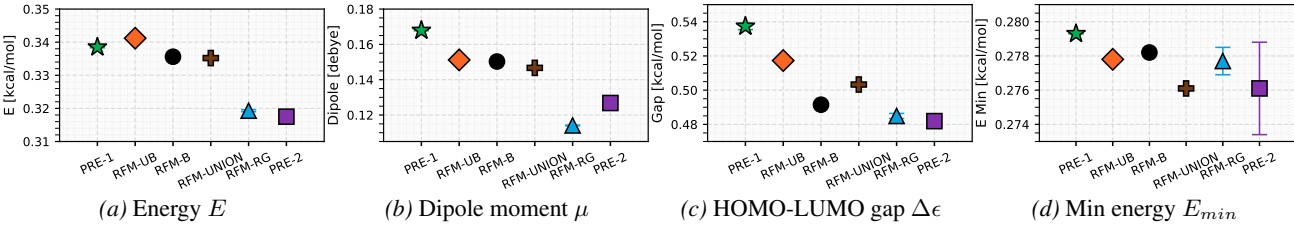

*Figure 5.* RFM can perform balanced (B), unbalanced (UB), reward-guided (RG) intersections, as well as unions (UNION) of prior ETFlow (Hassan et al., 2024) conformer generation models. We evaluate the resulting flow models in terms of median absolute errors of energy (5a), dipole moment (5b), HOMO–LUMO gap (5c), and minimum energy (5d). These results demonstrate the ability of RFM to compute new flow models whose properties predictably interpolate those of the available pre-trained flows.

Grimme, 2021), and modeling reaction pathways for catalyst design (Schmid et al., 2025). Given two prior ETFlow models (i.e., PRE-1 and PRE-2) (Hassan et al., 2024) with different property profiles, and we aim to merge them into a conformer generator whose profiles controllably interpolate between, or slightly improve upon their initial energetic ensemble property profile. In particular, we evaluate errors in energy, dipole moment, HOMO-LUMO gap and minimum energy of the generated structure ensemble compared to the equilibrium ensemble.

We run RFM initialized from PRE-2, to compute its balanced (B), unbalanced (UB), reward-guided (RG) intersection, and union variants. Figure 5a shows that the median absolute error (MAE) on the total energy $E$ interpolates between PRE-1 and PRE-2, RFM-B and RFM-UNION achieve intermediate errors of $\approx 0.3356$ and $0.3352$ kcal/mol as expected. On the other hand, the reward-guided (i.e., via energy minimization) counterpart reaches lower energy values, namely $\approx 0.3193$ kcal/mol, and the unbalanced variant ($\alpha_1 = 0.7, \alpha_2 = 0.3$) remains near PRE-1 at 0.3412 kcal/mol. These results validate the ability of RFM to perform unbalanced and reward-guided intersection. We report similar results for the dipole moment $\mu$ in Fig. 5b, for the HOMO–LUMO gap $\Delta\epsilon$ 5c, and minimum energy $E_{\min}$ 5d. Our evaluation indicates that RFM can perform (reward-guided) flow merging with conformer generation models, leading to flows with controllable interpolations or improvements over property profiles of pre-trained models.

Moreover, in Apx. F, we briefly investigate the computational cost of Reward-Guided Flow Merging, and report further experimental analysis on molecular and conformer design settings in Apx. H. Ultimately, although this work is primarily motivated by scientific discovery problems, we report in Apx. I an illustrative application of RFM to image-generation diffusion models.

## 8. Related Work

**Flow models fine-tuning via optimal control.** Several works have framed fine-tuning of flow and diffusion models to maximize expected reward functions under KL

regularization as an entropy-regularized optimal control problem (e.g., Uehara et al., 2024a; Tang, 2024; Uehara et al., 2024b; Domingo-Enrich et al., 2024; Gutjahr et al., 2026). More recently, De Santi et al. (2025a) introduced a framework for distributional fine-tuning, further specialized for CVaR optimization by Wang et al. (2026). The problem tackled in this work (see Eq. 5) instead extends the orthogonal setting of expected rewards with arbitrary divergences to the case with $n > 1$ pre-trained models. This generalization $(i)$ unifies flow control and merging, $(ii)$ renders possible to use of scalable control-based or RL schemes (e.g., Domingo-Enrich et al., 2024) for flow merging, and $(iii)$ enables reward-guided flow merging.

**Flow model merging and inference-time composition.** Recent works in inference-time flow and diffusion model composition introduced theory-backed schemes (e.g., Skreta et al., 2024; Bradley et al., 2025; Du et al., 2023). On the other hand, our work tackles the problem of (reward-guided) flow merging (e.g., Song et al., 2023), a significantly less explored research problem. Crucially, while inference-time flow composition aims to compose models at sampling time, flow merging aims to combine multiple flow models into one, and then disregard the prior models. This work provides a formal probability-space viewpoint on the latter problem, introduces interpretable merging operators (see Sec. 3) for highly expressive compositions (e.g., via generative circuits), provably implemented by RFM, which is to our knowledge the first scheme for provable reward-guided flow merging. Moreover, to our knowledge, the theoretical guarantees in Sec. 6 are first-of-their-kind for merging of flow and diffusion models. In particular, specializing them to specific operators e.g., intersection, yields highly relevant results, such as generative models safety guarantees via intersection with prior safe models.

**Convex and general utilities reinforcement learning.** Convex and General (Utilities) RL (Hazan et al., 2019; Zahavy et al., 2021; Zhang et al., 2020) generalizes RL to maximization of a concave (Hazan et al., 2019; Zahavy et al., 2021) or general (Zhang et al., 2020; Barakat et al., 2023) functional of the state distribution induced by a policy over a dynamical system's state space. Recent works tackled

the finite samples budget setting (e.g., Mutti et al., 2022b;a; De Santi et al., 2024). Similarly to previous optimization schemes for diffusion models (De Santi et al., 2025b;a; 2026), our framework (in Eq. 5) is related to Convex and General RL, with $p_1^\pi$ being the state distribution induced by policy $\pi$ over a subset of the flow process state space.

**Optimization over probability measures via mirror flows.** Recently, there has been a growing interest in devising theoretical guarantees for probability-space optimization problems in diverse fields of application. These include optimal transport (Aubin-Frankowski et al., 2022; Léger, 2021; Karimi et al., 2024), kernelized methods (Dvurechensky & Zhu, 2024), GANs (Hsieh et al., 2019), and manifold exploration (De Santi et al., 2025b; 2026) among others. To our knowledge, we present the first use of this theoretical framework to establish guarantees for flow and diffusion models merging.

## 9. Conclusion

We introduce a probability-space optimization framework for reward-guided flow merging, unifying and generalizing existing formulations. This allows to express diverse practically relevant operators over generative model densities (e.g., intersection, union, interpolation, logical expressions, and their reward-guided counterparts). We propose Reward-Guided Flow Merging, a mirror-descent scheme reducing complex merging tasks to sequential standard fine-tuning steps, solvable by established methods. Leveraging advances in mirror flows theory, we provide first-of-their kind guarantees for (reward-guided) flow merging. Empirical results on interpretable settings, molecular design, and conformer generation tasks demonstrate that our approach can steer pre-trained models to implement reward-guided merging tasks of high practical relevance.

## Acknowledgments

This publication was made possible by the ETH AI Center doctoral fellowship to Riccardo De Santi. The project has received funding from the Swiss National Science Foundation under NCCR Catalysis (grant number 180544 and 225147) and NCCR Automation (grant agreement 51NF40 180545), a National Centre of Competence in Research funded by the Swiss National Science Foundation. This work was supported by an ETH Zurich Research Grant.

## Impact Statement

This paper presents work whose goal is to advance the field of machine learning. There are many potential societal consequences of our work, none of which we feel must be specifically highlighted here.

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

# A. Appendix

# Contents

# B. Proofs for Section 6

## B.1. Proof of Theorem 6.1

**Stochastic Optimal Control.** We consider stochastic optimal control (SOC), which studies the problem of steering a stochastic dynamical system to optimize a specified performance criterion. Formally, let $(X_t^u)_{t \in [0,1]}$ be a controlled stochastic process satisfying the stochastic differential equation (SDE)

$$\mathrm{d}X_t^u = b(X_t^u, t)\,\mathrm{d}t + \sigma(t)\,u(X_t^u, t)\,\mathrm{d}t + \sigma(t)\,\mathrm{d}B_t, \qquad X_0^u \sim p_0, \tag{24}$$

where $u \in \mathcal{U}$ is an admissible control and $B_t$ is standard Brownian motion. The objective is to select $u$ to minimize the cost functional

$$\mathbb{E}\left[\int_0^1 \frac{1}{2}\|u(X_t^u, t)\|^2\,dt - g(X_1^u)\right], \tag{25}$$

where $\frac{1}{2}\|u(\cdot, t)\|^2$ represents the running cost and $g$ is a terminal reward. A standard application of Girsanov's theorem shows that Equation (25) is equivalent to the mirror descent iterate in Equation (20) with $\delta\mathcal{G}(p_1^{\pi_k}) \leftarrow g$ and $p_0 \leftarrow p^{pre}$ (Tang, 2024). In addition, it is well-known that in the context of diffusion-based generative modeling, the corresponding uncontrolled dynamics

$$\mathrm{d}X_t = -b(X_t, t)\,\mathrm{d}t + \sigma(t)\,\mathrm{d}B_t \tag{26}$$

coincide with the forward noising process used in score-based models (Song et al., 2021; Domingo-Enrich et al., 2024).

**Proof of Theorem 6.1.**

**Theorem 6.1** (SOC Retains Score Information). *Suppose that the prior diffusion model forward process converges to a standard Gaussian noise[1]. Then, the model returns by a SOC fine-tuning solver is such that:*

$$u^\star(x, t) := \sigma(t)\,\nabla \log p_t^k(x) \tag{19}$$

*where $p_t^k$ denotes the marginal distribution of the prior forward process, initialized at $p_1^{\pi_k}$, and $u^\star(x, t)$ the applied optimal control (see Sec. B). In other words,* REWARDGUIDEDFTSOLVER *exactly recovers the score.*

*Proof.* **Step 1.** Let $\mathbf{Q}^\star$ denote the optimal process solving Equation (24). A standard application of Girsanov's theorem shows that $\mathbf{Q}^\star$ also solves the *Schrödinger bridge problem*

$$\min_{\substack{\mathbf{Q}_0 = p^{\mathrm{pre}} \\ \mathbf{Q}_1 = \mathbf{Q}_1^\star}} D_{\mathrm{KL}}\big(\mathbf{Q} \,\|\, \mathbf{P}\big), \tag{27}$$

where $\mathbf{P}$ is the law of the uncontrolled dynamics

$$\mathrm{d}X_t = b(X_t, t)\,\mathrm{d}t + \sigma(t)\,\mathrm{d}B_t.$$

This equivalence holds because the SOC cost in Equation (24) penalizes control energy in the same way that Girsanov's theorem expresses a controlled SDE as a relative entropy with respect to its uncontrolled counterpart.

**Step 2.** Define the *forward process* $\mathbf{P}_{\mathrm{forward}}$ by

$$\mathrm{d}X_t = -b(X_t, t)\,\mathrm{d}t + \sigma(t)\,\mathrm{d}B_t. \tag{28}$$

By assumption, this process maps any initial distribution to the standard Gaussian at $t = 1$. In particular, starting from $X_0 \sim \mathbf{Q}_1^\star$, we obtain $X_1 \sim p^{\mathrm{pre}} = \mathcal{N}(0, I)$.

**Step 3.** Consider the *time-reversed Schrödinger bridge problem*

$$\min_{\substack{\overleftarrow{\mathbf{Q}}_0 = \mathbf{Q}_1^\star \\ \overleftarrow{\mathbf{Q}}_1 = p^{\mathrm{pre}}}} D_{\mathrm{KL}}\big(\overleftarrow{\mathbf{Q}} \,\|\, \mathbf{P}_{\mathrm{forward}}\big), \tag{29}$$

---

[1]This is a standard assumption in diffusion modeling

and denote its solution by $\overleftarrow{\mathbf{Q}}^\star$. Since relative entropy is invariant under bijective mappings and time-reversal is bijective, the optimizers of Equation (27) and Equation (29) satisfy

$$\overleftarrow{\mathbf{Q}}^\star = \overleftarrow{\mathbf{Q}^\star}$$

i.e., the optimal reversed bridge is simply the time-reversal of the forward bridge.

By **Step 2**, the process

$$\mathrm{d}X_t = -b(X_t, t)\,\mathrm{d}t + \sigma(t)\,\mathrm{d}B_t, \qquad X_0 \sim \mathbf{Q}_1^\star \tag{30}$$

solves Equation (29), achieving the minimum relative entropy (zero) while satisfying the prescribed marginals. Thus, invoking the relation $\overleftarrow{\mathbf{Q}}^\star = \overleftarrow{\mathbf{Q}^\star}$, the solution to Equation (27)—and hence to the SOC problem Equation (24)—is given by the time-reversal of Equation (30).

Finally, applying the classical time-reversal formula (Anderson, 1982) yields that $\mathbf{Q}^\star$ is given by

$$\mathrm{d}X_t = \left( b(\overleftarrow{X}_t, t) + \sigma^2(t)\,\nabla \log p_t(X_t) \right) \mathrm{d}t + \sigma(t)\,\mathrm{d}B_t,$$

where $p_t$ is the marginal density of Equation (30). Hence, REWARDGUIDEDFTSOLVER exactly recovers the score function. $\qquad\square$

### B.2. Rigorous Statement and Proof of Theorem 6.2

To prepare for the convergence analysis, we impose a few auxiliary assumptions. These assumptions are standard in the study of stochastic approximation and gradient flows, and typically hold in practical situations. Our proof strategy follows ideas that have also been employed in related works (De Santi et al., 2025b;a).

We begin with the entropy functional defined on probability measures:

$$\mathcal{H}(p) := \int p \log p. \tag{31}$$

In our analysis, $\mathcal{H}$ serves as the *mirror map* or *distance-generating function* (Mertikopoulos et al., 2024; Hsieh et al., 2019). The first condition addresses the behavior of the corresponding dual variables.

**Assumption B.1** (Precompactness of Dual Iterates). *The sequence of dual elements $\{\delta\mathcal{H}(p_1^{\pi_k})\}_k$ is precompact in the $L_\infty$ topology.*

This compactness property ensures that the interpolated dual trajectories remain confined to a bounded region of function space. Such a condition is crucial for invoking convergence results based on asymptotic pseudotrajectories. Variants of this assumption have appeared in the literature on stochastic approximation and continuous-time embeddings of discrete algorithms (Benaïm, 2006; Hsieh et al., 2019; Mertikopoulos et al., 2024).

**Assumption B.2** (Noise and Bias Conditions). *For the stochastic approximations used in the updates, we assume that almost surely:*

$$\|b_k\|_\infty \to 0, \tag{32}$$

$$\sum_k \mathbb{E}\left[ \gamma_k^2 \left( \|b_k\|_\infty^2 + \|U_k\|_\infty^2 \right) \right] < \infty, \tag{33}$$

$$\sum_k \gamma_k \|b_k\|_\infty < \infty. \tag{34}$$

These conditions, standard in the Robbins–Monro setting (Robbins & Monro, 1951; Benaïm, 2006; Hsieh et al., 2019), guarantee that the stochastic bias vanishes asymptotically while the cumulative noise remains under control. Together, they ensure that random perturbations do not obstruct convergence to the optimizer of the limiting objective.

With these assumptions in place, we can now state and prove the convergence guarantee.

**Theorem B.1** (Convergence guarantee in the trajectory setting). *Suppose Assumptions B.1–B.2 hold, and the step sizes $\{\gamma_k\}$ follow the Robbins–Monro conditions ($\sum_k \gamma_k = \infty$, $\sum_k \gamma_k^2 < \infty$). Then the sequence $\{p_1^{\pi_k}\}$ generated by Algorithm 1 converges almost surely, in the weak topology, to the optimum:*

$$p_1^{\pi_k} \rightharpoonup p_1^* \quad a.s., \tag{35}$$

*where $p_1^* = \mathbf{Q}_1^*$ for some $\mathbf{Q}^* \in \arg\max_{\mathbf{Q}:\mathbf{Q}_0 = p_0^{pre}} \mathcal{G}(\mathbf{Q}_1)$.*

*Proof.* We analyze the continuous-time mirror flow defined by

$$\dot{h}_t = \delta\mathcal{G}(p_1^t), \qquad p_1^t = \delta\mathcal{H}^\star(h_t), \tag{36}$$

where the Fenchel conjugate of $\mathcal{H}$ is given by $\mathcal{H}^\star(h) = \log \int e^h$ (Hsieh et al., 2019; Hiriart-Urruty & Lemaréchal, 2004).

To link the discrete dynamics to this continuous flow, we construct a piecewise linear interpolation of the iterates:

$$\hat{h}_t = h^{(k)} + \frac{t - \tau_k}{\tau_{k+1} - \tau_k}\big(h^{(k+1)} - h^{(k)}\big), \quad h^{(k)} = \delta\mathcal{H}(p_1^{\pi_k}), \quad \tau_k = \sum_{r=0}^{k} \alpha_r,$$

where $\{\alpha_r\}$ denotes the step-size sequence. This interpolation produces a continuous path $\hat{h}_t$ that tracks the discrete updates as the steps shrink.

Let $\Phi_u$ denote the flow map of equation 36 at time $u$. Standard results in stochastic approximation (Benaïm, 2006; Hsieh et al., 2019; Mertikopoulos et al., 2024) imply that for any fixed horizon $T > 0$, there exists a constant $C(T)$ such that

$$\sup_{0 \le u \le T} \|\hat{h}_{t+u} - \Phi_u(\hat{h}_t)\| \le C(T)\Big[\Delta(t-1, T+1) + b(T) + \gamma(T)\Big],$$

where $\Delta$ accounts for cumulative noise, $b$ for bias, and $\gamma$ for step-size effects. Under Assumptions B.1–B.2, these quantities vanish asymptotically, ensuring that $\hat{h}_t$ forms a precompact asymptotic pseudotrajectory (APT) of the mirror flow.

By the APT limit set theorem (Benaïm, 2006, Thm. 4.2), the limit set of a precompact APT is contained in the internally chain transitive (ICT) set of the underlying flow. In our case, Equation (36) corresponds to a gradient-like flow in the Hellinger–Kantorovich geometry (Mielke & Zhu, 2025), with $\mathcal{G}$ serving as a strict Lyapunov function. As $\mathcal{G}$ decreases strictly along non-stationary trajectories, the ICT set reduces to the collection of stationary points of $\mathcal{G}$.

Ultimately, notice that if $\mathcal{G}$ is composed of convex divergences (e.g., forward or reverse KL terms) possibly together with a linear component (i.e., any reward function $f$), its stationary points coincide with its global maximizers. Consequently, $\hat{h}_t$ converges almost surely to the set of maximizers of $\mathcal{G}$, which establishes the claim. Moreover, notice that within Sec. 6, we report the previously proved statement for stationary-points and then specialize it for global maximizers in the case of convex functionals. □

## C. Derivations of Gradients of First Variation

### C.1. A brief tutorial on first variation derivation

In this work, we focus on the functionals that are Fréchet differentiable: Let $V$ be a normed spaces. Consider a functional $F : V \to \mathbb{R}$. There exists a linear operator $A : V \to \mathbb{R}$ such that the following limit holds

$$\lim_{\|h\|_V \to 0} \frac{|F(f + h) - F(f) - A[h]|}{\|h\|_V} = 0. \tag{37}$$

We further assume that $V$ has enough structure such that every element of its dual (the space of bounded linear operator on $V$) admits a compact representation. For example, if $V$ is the space of bounded continuous functions with compact support, there exists a unique positive Borel measure $\mu$ with the same support, which can be identified as the linear functional. We denote this element as $\delta F[f]$ such that $\langle \delta F[f], h \rangle = A[h]$. Sometimes we also denote it as $\frac{\delta F}{\delta f}$. We will refer to $\delta F[f]$ as the first-order variation of $F$ at $f$.

In the following, we briefly present standard strategies to derive the first-order variation of two broad classes of functionals, including a wide variety of divergence measures, which can be employ to implement novel operators by Eq. 5. We consider: $(i)$ those defined in closed form with respect to the density (e.g., forward KL) and, $(ii)$ those defined via variational formulations (e.g., Wasserstein distance, reverse KL, and MMD).

- **Category 1: Functional defined in a closed form with respect to the density.** For this class of functionals, the first-order variations can typically be computed using its definition and chain rule.

  Recalling the definition of first variation (37), we can calculate the first-order variation of the mean functional, as a trivial example. Given a continuous and bounded function $r : \mathbb{R}^d \to \mathbb{R}$ and a probability measure $\mu$ on $\mathbb{R}^d$, define the functional $F(\mu) = \int r(x)\mu(x)dx$. Then we have:

$$|F(\mu + \delta\mu) - F(\mu) - \langle r, \delta\mu \rangle| = 0. \tag{38}$$

  Therefore we obtain that: $\delta F[\mu] = r$ for all $\mu$. In the following section, we compute similarly the first variation of the KL divergence.

- **Category 2: Functionals defined through a variational formulation.** Another fundamental subclass of functionals that plays a central role in this work is the one of functionals defined via a variational problem

$$F[f] = \sup_{g \in \Omega} G[f, g], \tag{39}$$

  where $\Omega$ is a set of functions or vectors independent of the choice of $f$, and $g$ is optimized over the set $\Omega$. We will assume that the maximizer $g^*(f)$ that reaches the optimal value for $G[f, \cdot]$ is unique (which is the case for the functionals considered in this project). It is known that one can use the Danskin's theorem (also known as the envelope theorem) to compute

$$\frac{\delta F[f]}{\delta f} = \partial_f G[f, g^*(f)], \tag{40}$$

  under the assumption that $F$ is differentiable (Milgrom & Segal, 2002).

### C.2. Derivation of First Variations used in Sec. 4

In the following, we derive explicitly the first variations employed in Sec. 1

- **Optimal transport and Wasserstein-p distance (Category 2)** Consider the optimal transport problem

$$\mathrm{OT}_c(u, v) = \inf_\gamma \left\{ \int \int c(x, y) d\gamma(x, y) : \int \gamma(x, y) dx = u(y), \int \gamma(x, y) dy = v(x) \right\} \tag{41}$$

  where

$$\Gamma = \left\{ \gamma : \int \gamma(x, y) dx = u(y), \int \gamma(x, y) dy = v(x) \right\}$$

It admits the following equivalent dual formulation

$$\mathrm{OT}_c(u,v) = \sup_{f,g} \left\{ \int f \, du + \int g \, dv : f(x) + g(y) \leq c(x,y) \right\} \tag{42}$$

By taking $c(x,y) = \|x-y\|^p$, we recover $\mathrm{OT}_c(u,v) = W_p(u,v)^p$. Let $\phi^*$ and $g^*$ be the solution to the above dual optimization problem. From the Danskin's theorem, we have

$$\frac{\delta}{\delta u} W_p(u,v)^p = \phi^*. \tag{43}$$

In the special case of $p = 1$, we know that $g^* = -\phi^*$ (note that the constraint can be equivalently written as $\|\nabla \phi\| \leq 1$), in which case $\phi^*$ is typically known as the critic in the Wasserstein-GAN framework (cf. Arjovsky et al., 2017).

- **Reverse KL divergence (Category 2)** We use the variational (Fenchel–Legendre) representation of the forward KL, $D_{KL}(p\|q)$, as in f-GAN (Nowozin et al., 2016):

$$D_{KL}(p\|q) = \sup_{\phi:\mathcal{X}\to\mathbb{R}} \left\{ \mathbb{E}_p \phi(x) - \mathbb{E}_q e^{\phi(x)-1} \right\} \tag{44}$$

which follows from the general f-divergence dual generator $f(u) = u \log u - u + 1$ whose conjugate is $f^*(t) = e^{t-1}$. For fixed $p$ and variable $q$, we define:

$$G(q,\phi) := \mathbb{E}_p \phi(x) - \mathbb{E}_q e^{\phi(x)-1} \tag{45}$$

Assuming uniqueness of a maximizer $\phi^*(p,q)$, Danskin's (or envelope) theorem yields the first variation by differentiating $G$ at $\phi^*$:

$$\frac{\delta}{\delta q(x)} D_{KL}(p\|q) = \frac{\delta}{\delta q(x)} \left( -\int q(x) e^{\phi^*(x)-1} du \right) = -e^{\phi^*(x)-1} \tag{46}$$

- **KL divergence (Category 1)** Consider the KL functional:

$$D_{KL}(p\|q) = -\int p \log \frac{p}{q}, dx \tag{47}$$

By the definition of the first-order variation (see Eq. 37), we have:

$$\delta D_{KL}(p\|q) = \log \frac{p}{q} + 1 \tag{48}$$

## D. Many-to-One Critics Reduction for Efficient FM

For the union operator, gradients are defined via critics $\{\phi_i^*\}_{i=1}^n$ learned with the standard variational form of reverse KL, as in f-GAN training of neural samplers (Nowozin et al., 2016). For $W_1$ interpolation, each $\phi_i^*$ plays the role of a Wasserstein-GAN discriminator with established learning procedures (Arjovsky et al., 2017). In both cases, each critic compares the fine-tuned density to a prior density $p_1^{pre,i}$, seemingly requiring one critic per prior. We prove that, surprisingly, this is unnecessary for the union operator, and conjecture that analogous results hold for other divergences.

**Proposition 1** (Union operator via pre-trained mixture density). *Let $\bar{p}_1^{pre} = \sum_{i=1}^n \alpha_i p_1^{pre,i} / \sum_{i=1}^n \alpha_i$, i.e., the $\alpha$-weighted mixture density of pre-trained models, then it holds that:*

$$\pi^* \in \arg\min_\pi \sum_{i=1}^n \alpha_i \, D_{KL}^R(p_1^\pi \,\|\, p_1^{pre,i}) \tag{49}$$

$$= \arg\min_\pi \left( \sum_{i=1}^n \alpha_i \right) D_{KL}^R(p_1^\pi \,\|\, \bar{p}_1^{pre}) \tag{50}$$

Prop. 1, which is proved in the following, implies that the union operator in Eq. 7 over $n$ prior models can be implemented by learning a single critic $\phi^*$, as shown in Sec. 7.

*Proof.* We prove the statement for $n = 2$, which trivially generalizes to any $n$. We first rewrite the LHS optimization problem as:

$$\arg\min_\pi \mathcal{F}(p^\pi) \tag{51}$$

where we denote $p_1^\pi$ by $p^\pi$ for notational concision and define $p_1 = p^{pre,i}$ and $p_2 = p^{pre,2}$. Then we have:

$$\mathcal{F}(p^\pi) = \alpha_1 \mathbb{E}_{p_1}[\log p_1 - \log p^\pi] + \alpha_2 \mathbb{E}_{p_2}[\log p_2 - \log p^\pi] \tag{52}$$

$$= \alpha_1 \mathbb{E}_{p_1} \log p_1 + \alpha_2 \mathbb{E}_{p_2} \log p_2 - \left( \alpha_1 \mathbb{E}_{p_1} \log p^\pi + \alpha_2 \mathbb{E}_{p_2} \log^\pi \right) \tag{53}$$

We now write the following, where $\bar{p}$ denotes $\bar{p}_1^{pre}$:

$$\mathbb{E}_{\bar{p}} \log p^\pi = \int \log p^\pi(x) \bar{p}(x) \, \mathrm{d}x \tag{54}$$

$$= \int \log p^\pi(x) \left[ \frac{\alpha_1 p_1}{\alpha_1 + \alpha_2} + \frac{\alpha_2 p_2}{\alpha_1 + \alpha_2} \right](x) \, \mathrm{d}x \tag{55}$$

$$= \frac{1}{\alpha_1 + \alpha_2} \left( \log p^\pi(x) \alpha_1 p_1(x) + \log p^\pi(x) \alpha_2 p_2(x) \right) \tag{56}$$

$$= \frac{1}{\alpha_1 + \alpha_2} \left( \alpha_1 \mathbb{E}_{p_1} \log p^\pi + \alpha_2 \mathbb{E}_{p_2} \log p^\pi \right) \tag{57}$$

By combining Eq. 53 and 57, we obtain:

$$\mathcal{F}(p^\pi) = \alpha_1 \mathbb{E}_{p_1} \log p_1 + \alpha_2 \mathbb{E}_{p2} \log p_2 - (\alpha_1 + \alpha_2) \mathbb{E}_{\bar{p}} \log p^\pi \tag{58}$$

Therefore,

$$\arg\min_\pi \mathcal{F}(p^\pi) = \arg\min_\pi \underbrace{\alpha_1 \mathbb{E}_{p_1} \log p_1 + \alpha_2 \mathbb{E}_{p_2} \log p_2}_{\text{constant}} - (\alpha_1 + \alpha_2) \mathbb{E}_{\bar{p}} \log p^\pi \tag{59}$$

$$= \arg\min_\pi -(\alpha_1 + \alpha_2) \mathbb{E}_{\bar{p}} \log p^\pi \tag{60}$$

$$= \arg\min_\pi -(\alpha_1 + \alpha_2) \mathbb{E}_{\bar{p}} \log p^\pi + \underbrace{(\alpha_1 + \alpha_2) \mathbb{E}_{\bar{p}} \log \bar{p}}_{\text{constant}} \tag{61}$$

$$= \arg\min_\pi (\alpha_1 + \alpha_2) D_{KL}(\bar{p} \| p^\pi) \tag{62}$$

$$\tag{63}$$

Which concludes the proof. $\qquad\square$

# E. Reward-Guided Flow Merging (RFM) Implementation

In the following, we provide an example of detailed implementations for REWARDGUIDEDFTSOLVER employed in Sec. 4 by Reward-Guided Flow Merging, as well as REWARDGUIDEDFTSOLVERRUNNINGCOSTS, leveraged in Sec. 5 to scalably implement the AND operator. While the oracle implementation we report for completeness for REWARDGUIDEDFTSOLVER corresponds to classic Adjoint Matching (AM) (Domingo-Enrich et al., 2024), the one for REWARDGUIDEDFTSOLVERRUN-NINGCOSTS trivially extends AM base implementation to account for the running cost terms introduced in Eq. 17.

## E.1. Implementation of REWARDGUIDEDFTSOLVER

Before detailing the implementations, we briefly fix notation. Both algorithms explicitly rely on the interpolant schedules $\kappa_t$ and $\omega_t$ from equation 1. In the flow-model literature, these are more commonly denoted $\alpha_t$ and $\beta_t$. We write $u^{\text{pre}}$ for the velocity field induced by the pre-trained policy $\pi^{\text{pre}}$, and $u^{\text{fine}}$ for the velocity field induced by the fine-tuned policy. In essence, each algorithm first draws trajectories and then uses them to approximate the solution of a surrogate ODE; its marginals serve as regression targets for the control policy (Section 5 Domingo-Enrich et al., 2024).

---

**Algorithm 2** REWARDGUIDEDFTSOLVERRUNNINGCOSTS via AM

---

1: **input:** Pre-trained FM velocity field $u^{\text{pre}}$, step size $h$, number of fine-tuning iterations $N$, gradient of reward $\nabla r$, fine-tuning strength $\eta_k$
2: **init:** Initialize fine-tuned vector fields: $u^{\text{finetune}} = u^{\text{pre}}$ with parameters $\theta$.
3: **for** $n \in \{0, \dots, N-1\}$ **do**
4:     Sample $m$ trajectories $\boldsymbol{X} = (X_t)_{t \in \{0,\dots,1\}}$ with memoryless noise schedule:

$$\sigma(t) = \sqrt{2\kappa_t \left( \frac{\dot{\omega}_t}{\omega_t} \kappa_t - \dot{\kappa}_t \right)} \tag{64}$$

5:     i.e.,:
$$X_{t+h} = X_t + h\left( 2u_\theta^{\text{finetune}}(X_t, t) - \frac{\dot{\omega}_t}{\omega_t} X_t \right) + \sqrt{h}\,\sigma(t)\,\varepsilon_t, \quad \varepsilon_t \sim \mathcal{N}(0, I), \quad X_0 \sim \mathcal{N}(0, I). \tag{51}$$

6:     For each trajectory, solve the *lean adjoint ODE* backwards in time from $t = 1$ to $0$, e.g.:

$$\tilde{a}_{t-h} = \tilde{a}_t + h\,\tilde{a}_t^\top \nabla_{X_t}\left( 2v^{\text{base}}(X_t, t) - \frac{\dot{\omega}_t}{\omega_t} X_t \right), \quad \tilde{a}_1 = \eta_k \nabla r(X_1). \tag{52}$$

7:     Note that $X_t$ and $\tilde{a}_t$ should be computed without gradients, i.e.,

$$X_t = \texttt{stopgrad}(X_t) \tag{65}$$
$$\tilde{a}_t = \texttt{stopgrad}(\tilde{a}_t) \tag{66}$$

8:     For each trajectory, compute the following Adjoint Matching objective:

$$\mathcal{L}_{\text{Adj-Match}}(\theta) = \sum_{t \in \{0,\dots,1-h\}} \left\| \frac{2}{\sigma(t)} \left( v_\theta^{\text{finetune}}(X_t, t) - u^{\text{base}}(X_t, t) \right) + \sigma(t)\,\tilde{a}_t \right\|^2. \tag{53}$$

9:     Compute the gradient $\nabla_\theta \mathcal{L}(\theta)$ and update $\theta$ using favorite gradient descent algorithm.
10: **end for**
**Output:** Fine-tuned vector field $v^{\text{finetune}}$

---

## E.2. Implementation of REWARDGUIDEDFTSOLVERRUNNINGCOSTS

The following REWARDGUIDEDFTSOLVERRUNNINGCOSTS is algorithmically identical to REWARDGUIDEDFTSOLVERRUN-NINGCOSTS, with the only difference that the lean adjoint computation now integrates a running-cost term $f_t$, defined as follows (see Sec. 5):

$$f_t(x) := \delta \left( \sum_{i=1}^{n} \alpha_i\, D_{KL}(p_t^\pi \,\|\, p_t^{pre,i}) \right)(x, t), \quad t \in [0, 1) \tag{67}$$

---

**Algorithm 3** REWARDGUIDEDFTSOLVERRUNNINGCOSTS via AM with running costs

---

1: **input:** Pre-trained FM velocity field $v^{\text{base}}$, step size $h$, number of fine-tuning iterations $N$, $f_t = \nabla \delta \mathcal{G}_t(p_t^{\pi^k})$, weight $\gamma_k$, weight schedule $\lambda$

2: Initialize fine-tuned vector fields: $v^{\text{finetune}} = v^{\text{base}}$ with parameters $\theta$.

3: **for** $n \in \{0, \dots, N-1\}$ **do**

4:     Sample $m$ trajectories $\boldsymbol{X} = (X_t)_{t \in \{0, \dots, 1\}}$ with memoryless noise schedule:

$$\sigma(t) = \sqrt{2\kappa_t \left( \frac{\dot{\omega}_t}{\omega_t} \kappa_t - \dot{\kappa}_t \right)} \tag{68}$$

5:     i.e.,:

$$X_{t+h} = X_t + h \left( 2 v_\theta^{\text{finetune}}(X_t, t) - \frac{\dot{\omega}_t}{\omega_t} X_t \right) + \sqrt{h}\, \sigma(t)\, \varepsilon_t, \quad \varepsilon_t \sim \mathcal{N}(0, I), \quad X_0 \sim \mathcal{N}(0, I). \tag{40}$$

6:     For each trajectory, solve the *lean adjoint ODE* backwards in time from $t = 1$ to $0$, e.g.:

$$\tilde{a}_{t-h} = \tilde{a}_t + h\, \tilde{a}_t^\top \nabla_{X_t} \left( 2 v^{\text{base}}(X_t, t) - \frac{\dot{\omega}_t}{\omega_t} X_t \right) - h \gamma_k \lambda_t f_t(X_t) \tag{69}$$

$$\tilde{a}_1 = -\gamma_k \lambda_1 \nabla_{X_1} \delta \mathcal{G}_1(p_1^{\pi^k})(X_1). \tag{41}$$

7:     Note that $X_t$ and $\tilde{a}_t$ should be computed without gradients, i.e.,

$$X_t = \texttt{stopgrad}(X_t) \tag{70}$$

$$\tilde{a}_t = \texttt{stopgrad}(\tilde{a}_t) \tag{71}$$

8:     For each trajectory, compute the Adjoint Matching objective:

$$\mathcal{L}_{\text{Adj-Match}}(\theta) = \sum_{t \in \{0, \dots, 1-h\}} \left\| \frac{2}{\sigma(t)} \left( v_\theta^{\text{finetune}}(X_t, t) - v^{\text{base}}(X_t, t) \right) + \sigma(t)\, \tilde{a}_t \right\|^2. \tag{}$$

9:     Compute the gradient $\nabla_\theta \mathcal{L}(\theta)$ and update $\theta$ using a gradient descent step

10: **end for**

**Output:** Fine-tuned vector field $u^{\text{finetune}}$

---

# F. Reward-Guided Flow Merging (RFM): Computational Complexity, Cost, and Approximate Fine-Tuning Oracles

Reward-Guided Flow Merging (RFM, see Alg. 1) is a sequential fine-tuning scheme which, at each of the $(K)$ outer iterations, calls a reward-guided fine-tuning oracle such as REWARDGUIDEDFTSOLVER (see Apx. E.2). In practice, each oracle call performs $(N)$ gradient steps of Adjoint Matching (see Apx. E.2). At first sight, this suggests that the computational complexity of RFM scales linearly in $K$ with respect to a standard fine-tuning run with $(N)$ steps. However, this worst-case view does not fully capture the practical computational cost. We highlight two observations.

**Approximate fine-tuning oracle.**    First, RFM can operate reliably with a rather *approximate fine-tuning oracle*, i.e., with relatively small values of $(N)$. We evaluate this phenomenon by replicating the objective curve of Fig. 2d with same parameters and setting, for three different configurations of $(K, N)$ that keep the total budget $(K \cdot N = 300)$ fixed but vary the outer (i.e., $K$) and inner (i.e., $N$) iteration counts:

- $K = 10, ; N = 30$

- $K = 15, ; N = 20$ (as in Fig. 2d)

- $K = 30, ; N = 10$

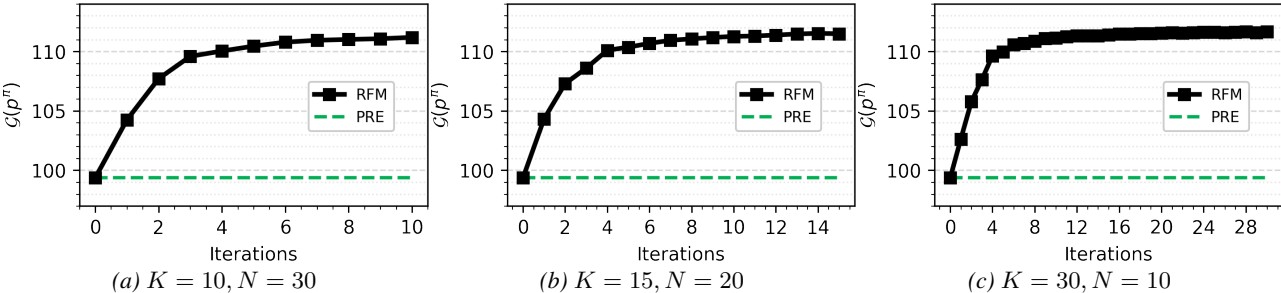

*(a) $K = 10, N = 30$*      *(b) $K = 15, N = 20$*      *(c) $K = 30, N = 10$*

*Figure 6.* (left) RFM run for reward-guided intersection with $K = 10, N = 30$, (center) RFM run for reward-guided intersection with $K = 15, N = 20$, (right) RFM run for reward-guided intersection with $K = 30, N = 10$.

The three corresponding curves are reported in Fig. 6. Empirically, all three settings achieve nearly identical final objective values, indicating that a more approximate oracle (smaller $(N)$) can be compensated by increasing the number of outer RFM iterations $(K)$, and vice versa, as long as the total optimization budget remains comparable. We observe a similar behaviour also on real-world, higher-dimensional, experiments (see Sec. 7 and Apx. I), where we values of $K$ vary from $K = 1$ to $K = 37$.

$K/N$ **Trade-off.**    Second, the runtimes of these configurations are of the same order. On our implementation, the runs with $((K, N) = (10, 30), (15, 20), (30, 10))$ require approximately 1615 s, 1643 s, and 1870 s, respectively, showing a very light increase depending on $K$. This further supports the view that practitioners can trade off a cheaper but less accurate inner oracle (small $(N)$) against a slightly larger number of outer RFM steps (larger $(K)$), and vice versa, without incurring prohibitive additional cost. Since RFM effectively solves a convex/non-convex optimization problem in probability space, we believe that classic convex optimization provides an interpretable framework for trading-off $N$ and $K$, by interpreting $N$ as the typical step-size, or learning rate, and $K$ as the typical number of gradient steps. Clearly, higher learning rates typically require less gradient steps and vice versa. Ultimately, one should notice that increasing $N$ does not directly imply better solution quality of the fine-tuning oracle, as it is the case for the oracle we employ within Sec. 7 (i.e., Adjoint Matching (Domingo-Enrich et al., 2024)), for which performance can degrade for excessively high values of $N$.

# G. Experimental Details

## G.1. Illustrative Examples Experimental Details

Numerical values in all plots shown within Sec. 7 are means computed over diverse runs of RFM via 5 different seeds. Error bars correspond to 95% Confidence Intervals.

**Shared experimental setup.** For all illustrative experiments we utilize Adjoint Matching (AM) [14 ] for the entropy-regularized fine-tuning solver in Algorithm 1. Moreover, the stochastic gradient steps within the AM scheme are performed via an Adam optimizer.

**Intersection Operator.** The balanced plot (see Fig. 2b is obtained by running RFM with $\alpha = [0.1, 0.1]$, for $K = 80$ iterations, $\gamma_k = 28$, and $\lambda_t = 0.2$ for $t > 1 - 0.05$, and $\lambda_t = 0.4$ otherwise.

For the balanced, reward-guided case in Fig. 2c, we consider a reward function that is maximized by increasing the $x_2$ coordinate. We run RFM with $\alpha = [0.1, 0.1]$, for $K = 15$ iterations, $\gamma_k = 1.2$, and $\lambda_t = 0.2$ for $t > 1 - 0.05$, and $\lambda_t = 0.4$ otherwise.

**Union Operator.**

In both cases, we learn a critic via standard f-GAN (Nowozin et al., 2016) with 300 gradient steps at each iteration $k \in [K]$ and continually fine-tune the same critic over subsequent iterations. For critic learning, we use a learning rate of $5 \exp(-5)$.

For the balanced case, in Fig. 2f, we run RFM with $\alpha = [1.0, 1.0]$. We use $K = 13$ iterations, $\gamma_k = 0.001$.

For the unbalanced case in Fig. 2g, we run RFM with $\alpha = [0.2, 1.8]$. Notice that up to normalization this is equivalent to $[0.1, 0.9]$ as reported in Fig. 2g for the sake of interpretability. We use $K = 13$ iterations, $\gamma_k = 0.001$.

**Interpolation Operator.** In both cases, we learn a critic via standard f-GAN (Nowozin et al., 2016) with 800 gradient steps at each iteration $k \in [K]$ and continually fine-tune the same critic over subsequent iterations. For critic learning, we use a learning rate of $1 \exp(-5)$, and gradient penalty of $10.0$ to enforce 1-Lip. of the learned critic.

For the case where $\pi^{init} := \pi^{pre,1}$ (i.e., left pre-trained model), in Fig. 2j, we run RFM with $\alpha = [1.0, 1.0]$. We use $K = 6$ iterations, $\gamma_k = 1.0$.

For the case where $\pi^{init} := \pi^{pre,2}$ (i.e., right pre-trained model), in Fig. 2k, we run RFM with $\alpha = [1.0, 1.0]$. We use $K = 6$ iterations, $\gamma_k = 1.0$.

**Complex Logic Expressions via Generative Circuits.** Pre-trained flows $\pi_1$ and $\pi_2$, as well as $\pi_1$ and $\pi_2$ are intersected via RFM with $\gamma_k = 1$, for $K = 20$, and $\lambda_t = 0.1$. The union operator is implemented with $K = 30$, $\gamma_k = 0.0009$, 300 critic steps and learning rate $5 \exp(-5)$.

## G.2. Molecular Design Case Study

Our base model FlowMol2 CTMC (i.e., PRE-1) (Dunn & Koes, 2024) is pretrained on the GEOM-Drugs dataset (Axelrod & Gomez-Bombarelli, 2022). We obtain our second model (i.e., PRE-2) by finetuning PRE-1 with AM (Domingo-Enrich et al., 2024) to generate poses with lower single point total energy wrt. the continuous atomic positions as calculated with dxtb at the GFN1-xTB level of theory (Friede et al., 2024). We then run RFM with $K = 50$, $\gamma = 0.001$ for the balanced flow merging, and $K = 20$, $\gamma = 0.005$ to obtain the unbalanced flow merging. For reward-guided flow merging (RFM-RG), we set $\gamma = 0.1$ and obtain the best model after $K = 11$. All models start from PRE-1, i.e., $\pi^{init} = \pi^{pre,1}$. All results for merging pre-trained models on GEOM can be found in Table 1. Running RFM-RG with $\alpha = 3$ and $\gamma = 0.001$, we obtain a model after $K = 35$ that keeps the validity of its base models while implementing the reward-guided intersection. We note that beyond validity, a critical step towards practical application will be to integrate molecular stability and synthesizability. Our RFM formulation straightforwardly supports these extensions in the reward functional, and we leave their implementation to future work. For our second case-study - the OR operator - we use FlowMol2 CTMC pre-trained on QM9 (Ramakrishnan et al., 2014). We limit dimensionality to reduce the problem complexity by sampling 10 atoms per molecule, and run RFM with $\gamma = 100$, $K = 37$. In particular Figure 7 shows that the estimated mean of the model $\pi^*$ obtained via RFM matches the average total energy of $\pi^{pre,1}$ and $\pi^{pre,2}$ as predicted by the closed-form solution for the union operator presented in Sec. 3. In Fig. 7, OR denotes the final policy $\pi^*$ returned by RFM.

| Model | Mean total energy [Ha] | Mean validity [%] |
|---|---|---|
| PRE-1 | $-8.09 \pm 0.31$ | $76.44 \pm 1.7$ |
| RFM-B | $-10.95 \pm 0.28$ | $74.34 \pm 0.9$ |
| RFM-RG | $-12.85 \pm 0.16$ | $74.02 \pm 1.18$ |
| RFM-UB | $-13.69 \pm 0.28$ | $72.78 \pm 0.4$ |
| PRE-2 | $-14.76 \pm 0.29$ | $68.04 \pm 0.8$ |

*Table 1.* Mean total energy and validity with standard deviation, averaged over 5 different seeds. Suffixes: B - balanced ; UB - unbalanced; RG - reward-guided flow merging

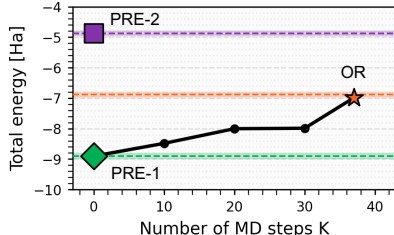

*Figure 7.* Union on QM9

### G.3. Conformer Generation Case Study

We finetune the GEOM-QM9 pre-trained ETFlow model (denoted PRE-1) with AM on the molecular system `C#C[C@H](C=O)CCC` to obtain PRE-2, using the same total energy objective as in the molecular design case study. This is also the molecular system we perform our evaluations on. For the subsequent merging experiments, we choose the lower-energy PRE-2 as the base model, i.e., $\pi^{init} = \pi^{pre,2}$. Balanced merging is performed with $\alpha_1 = \alpha_2 = 1$, $\gamma = 0.025$ and $K = 6$. The unbalanced merging is run with $\alpha_1 = 0.7$ and $\alpha_2 = 0.3$ and we take the model after $K = 8$ steps with $\gamma = 5e - 5$. The reward-guided merging model was obtained with $\gamma = 0.025$ after $K = 6$, and the union model after $K = 1$ with $\gamma = 1e - 3$ and critics with the same GNN backbone as ETFlow. We show all results for the conformer generation case study in Tab. 2

| Model | $E$ [kcal/mol] | $\mu$ [debye] | $\Delta\epsilon$ [kcal/mol] | $E_{min}$ [kcal/mol] |
|---|---|---|---|---|
| PRE-1 | $0.3385 \pm 0.0002$ | $0.1679 \pm 0.0002$ | $0.5373 \pm 0.0019$ | $0.2793$ |
| RFM-UB | $0.3412$ | $0.1512$ | $0.5173$ | $0.2778$ |
| RFM-B | $0.3356 \pm 0.0001$ | $0.1503 \pm 0.0002$ | $0.4915 \pm 0.0014$ | $0.2782$ |
| RFM-UNION | $0.3352$ | $0.1467$ | $0.5033$ | $0.2761$ |
| RFM-RG | $0.3193 \pm 0.0003$ | $0.1141 \pm 0.0002$ | $0.4849 \pm 0.0015$ | $0.2777 \pm 0.0008$ |
| PRE-2 | $0.3175 \pm 0.0006$ | $0.1268 \pm 0.0006$ | $0.4819 \pm 0.0010$ | $0.2761 \pm 0.0027$ |

*Table 2.* Median Absolute Errors for energy $E$, dipole moment $\mu$, HOMO-LUMO gap $\Delta\epsilon$, and minimum energy $E_{min}$ across different models. We report mean and standard deviation over 5 different seeds.

## H. Further Experimental Analysis on Molecular and Conformer Design Tasks

In this section, we report additional experimental analyses complementing the molecular design and conformer generation results in Sec. 7. These experiments further assess whether RFM can merge models with different architectures, how it compares to simple weight-space averaging, whether reward-guided merging preserves molecular diversity, and whether conformer generation results extend beyond a single molecular system.

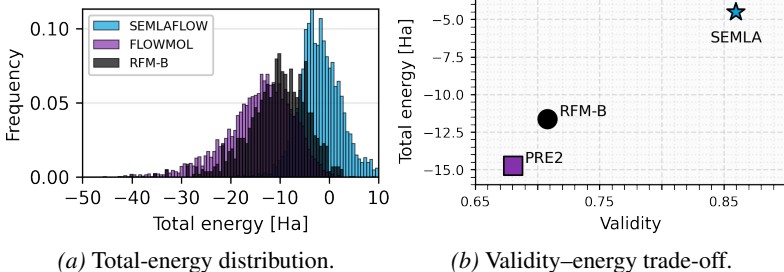

*(a)* Total-energy distribution.   *(b)* Validity–energy trade-off.

*Figure 8.* RFM-B merging of the FlowMol PRE-2 model from our molecular design case study and a SemlaFlow model on GEOM-Drugs. The result illustrates that RFM can merge molecular generators with different architectures, unlike weight-space averaging methods.



| | Recall | | | | Precision | | | |
|---|---|---|---|---|---|---|---|---|
| | Coverage ↑ | | AMR ↓ | | Coverage ↑ | | AMR ↓ | |
| | mean | median | mean | median | mean | median | mean | median |
| PRE-1 | 0.941 | 1.0 | 0.096 | 0.041 | 0.906 | 1.0 | 0.130 | 0.071 |
| RFM-B | 0.941 | 1.0 | 0.096 | 0.040 | 0.906 | 1.0 | 0.130 | 0.071 |
| PRE-2 | 0.930 | 1.0 | 0.173 | 0.130 | 0.892 | 1.0 | 0.211 | 0.153 |

*(a)* Weight-space averaging baseline.          *(b)* QM9 conformer generation metrics.

*Figure 9.* Additional baselines and conformer-generation evaluation. **(a)** Weight-space averaging on the illustrative two-dimensional setting: left, two partially overlapping base models as in Fig. 2a; right, the model obtained via weight averaging, which does not implement a controlled density-level merging operation. **(b)** Coverage and Average Minimum RMSD (AMR) metrics for the base ETFlow models and the merged RFM-B conformer generation model on the QM9 test set. Higher coverage and lower AMR are better.

## H.1. Weight-Space Averaging and Architecture-Agnostic Merging

**Comparison to weight-space averaging.**   We compare RFM against a simple weight-space averaging baseline, motivated by model-soup (Wortsman et al., 2022) and diffusion-soup (Biggs et al., 2024) approaches. Since such methods require compatible architectures, we first evaluate them in the illustrative two-dimensional setting from Fig. 2a, where the two prior models partially overlap and can in principle be averaged in weight space. As shown in Fig. 9a, weight averaging does not recover a meaningful density-level operation: the averaged model fails to represent either the intended intersection or union of the base distributions, and instead yields a degenerate density. This illustrates the main conceptual distinction: RFM optimizes an explicit distributional objective, whereas weight averaging provides no direct control over the density operator implemented by the merged model.

**Merging different molecular-generator architectures.**   A further limitation of weight-space merging is that all models must share the same architecture. RFM does not impose this restriction, since it operates through sampling and fine-tuning procedures rather than parameter alignment. To test this property, we merge the FlowMol PRE-2 model from the molecular design case study with a SemlaFlow model on GEOM-Drugs. We compute a balanced intersection model, denoted RFM-B, using $K = 4$ RFM iterations, step size $\gamma = 2 \cdot 10^{-4}$, solver learning rate $2 \cdot 10^{-4}$, and prior-divergence weight $\alpha = 2$, where the stronger KL regularization helps preserve high validity. As shown in Fig. 8, RFM-B combines the two heterogeneous priors: the merged model occupies an intermediate energy region while maintaining high validity. This supports the use of RFM as an architecture-agnostic model-merging procedure for molecular generation.

## H.2. Diversity of Merged Molecular Models

Reward-guided adaptation can in principle reduce diversity by concentrating samples in a narrow high-reward region. To assess whether this occurs in our molecular design experiments, we evaluate molecular diversity through Tanimoto similarities between generated molecules. We report both intra-model similarity, which measures diversity within the samples of a given model, and inter-model similarity, which compares generated molecules from each merged model to those from the corresponding base models.

As shown in Fig. 10, the intra-model similarity distributions of RFM-B, RFM-UB, and RFM-RG remain comparable to those of PRE-1 and PRE-2. Moreover, the inter-model similarities between merged and base models remain in the same

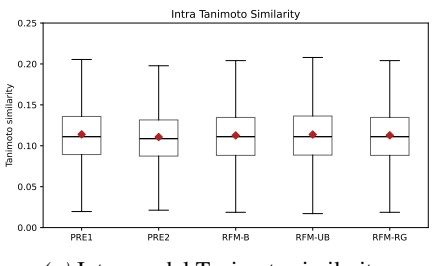 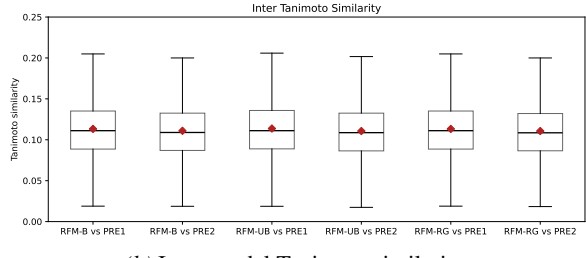

*(a)* Intra-model Tanimoto similarity.  *(b)* Inter-model Tanimoto similarity.

*Figure 10.* Diversity evaluation via Tanimoto similarity of generated molecules. Intra-model similarity measures diversity within each model's generated molecules, while inter-model similarity compares generated molecules from a merged model to those from its base models. The merged models exhibit no significant evidence of mode collapse.

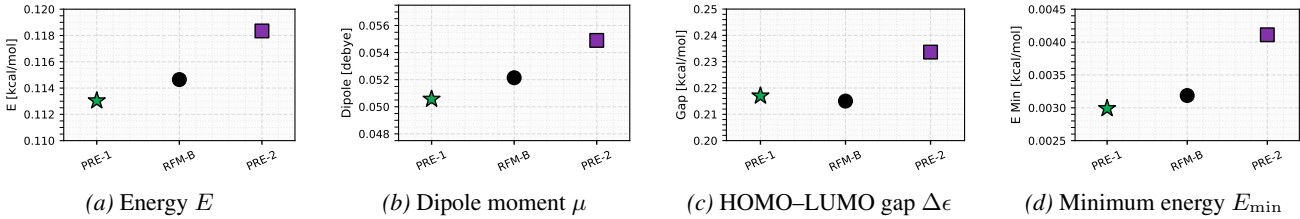

*(a)* Energy $E$  *(b)* Dipole moment $\mu$  *(c)* HOMO–LUMO gap $\Delta\epsilon$  *(d)* Minimum energy $E_{\min}$

*Figure 11.* Balanced RFM merging of two ETFlow conformer generation models on QM9. PRE-2 is obtained by fine-tuning PRE-1 on the QM9 validation set. RFM-B is also trained on the validation set, and all reported metrics are evaluated on the QM9 test set.

range, suggesting that the merged models do not collapse to a small set of highly similar molecules or simply duplicate one prior. These results support the interpretation that RFM preserves molecular diversity while implementing balanced, unbalanced, and reward-guided merging.

### H.3. Conformer Generation Beyond a Single Molecular System

The conformer generation case study in Sec. 7 evaluated RFM on a single molecular system. We therefore extend the evaluation to a broader QM9 conformer generation setting using ETFlow. We start from a pre-trained ETFlow model PRE-1 and obtain PRE-2 by fine-tuning PRE-1 with Adjoint Matching on the QM9 validation set, using learning rate $10^{-4}$ for 2 epochs. We then run balanced RFM merging on the QM9 validation set with $K = 2$, $\gamma = 10^{-3}$, and solver learning rate $10^{-5}$. All results are reported on the held-out QM9 test set.

Figure 11 reports errors for energy, dipole moment, HOMO–LUMO gap, and minimum energy. The merged model RFM-B closely matches the stronger base model PRE-1 and improves over PRE-2 across the reported properties. Fig. 9 further reports recall and precision coverage together with Average Minimum RMSD (AMR). RFM-B matches PRE-1 on mean recall coverage ($0.941$), mean precision coverage ($0.906$), and mean AMR, while outperforming PRE-2. These results indicate that balanced RFM merging can be applied beyond a single conformer system without degrading standard conformer-generation metrics.

## I. Beyond Molecules: Reward-Guided Flow Merging of Pre-Trained Image Models

We further showcase the capabilities of Reward-Guided Flow Merging on a small-scale, yet informative experiment for image generation. In the following, we consider pretrained CIFAR-10 image models (Krizhevsky, 2009) and use the LAION aesthetics predictor V1 (Schuhmann et al., 2022) as a reward model. Specifically, the aesthetics predictor was trained on a subset of the SAC dataset (Pressman et al., 2022) with available ratings from 1 (low preference / aesthetics) to 10 (high preference). The goal of this case study is to show that RFM can merge two models, PRE-1 and PRE-2, while optimizing the aesthetics score. We perform reward-guided flow merging with PRE-2 as the base model, obtaining the model RFM-RG after $K = 11$ iterations with $\gamma = 1$ and $\alpha_i = 1$. The numerical results in Tab. 3 show that RFM can successfully intersect multiple prior flow image models while maximizing the aesthetic score. In particular, the fine-tuned model achieves a score of $3.64 \pm 0.53$ against $3.16 \pm 0.66$ and $3.23 \pm 0.58$ of PRE-1 and PRE-2 respectively. We also report sample images of the discussed models in Fig. 12.

| Model | Mean aesthetic score |
|--------|---------------------|
| PRE-1 | $3.16 \pm 0.66$ |
| PRE-2 | $3.23 \pm 0.58$ |
| RFM-RG | $3.64 \pm 0.53$ |

*Table 3.* RFM can perform reward-guided (RG) intersections of pre-trained CIFAR-10 image models (Krizhevsky, 2009). We evaluate the resulting models in terms of mean aesthetic score (i.e., the reward) over 1000 samples, and report one std.

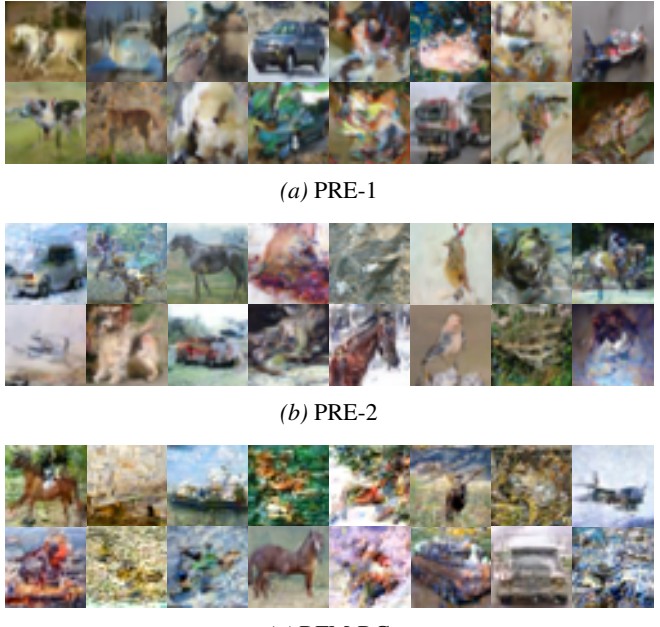

*(a)* PRE-1

*(b)* PRE-2

*(c)* RFM-RG

*Figure 12.* Images generated by the two pre-trained flow models (i.e., PRE-1, PRE-2), and by the flow model obtained via reward-guided intersection (i.e., RFM-RG).

