# OpenReview forum: "A Unified Density Operator View of Flow Control and Merging"
_ICML.cc/2026/Conference — ICML 2026 regular_

### Official Review · Reviewer_KLp5 · 2026-03-12

**Soundness:** 2
**Presentation:** 3
**Significance:** 3
**Originality:** 4
**Overall Recommendation:** 4
**Confidence:** 3

**Summary:**

This paper unifies reward-guided fine-tuning and model merging for continuous-time generative models into a single probability-space framework. By defining "density operators" (Intersection, Union, Interpolation), the authors can compose multiple pre-trained models into complex generative circuits. To optimize these non-linear objectives, they propose Reward-Guided Flow Merging, a mirror-descent algorithm that reduces the problem to sequential control-based fine-tuning steps. They additionally introduce a stable, process-level variant for the intersection operator to bypass score singularities at the terminal boundary. RFM is evaluated on 2D illustrative settings, molecular design and conformer generation

**Compliance With Llm Reviewing Policy:**

Affirmed.

**Final Justification:**

The paper offers an original and technically solid framework for unifying reward-guided fine-tuning and model merging in continuous-time generative models, supported by elegant formulations and encouraging results on both toy and scientific tasks. Its main limitations are that the theoretical guarantees depend on idealized assumptions that remain difficult to verify in practical neural settings, and the initial submission did not fully address diversity preservation under reward guidance. The rebuttal was helpful and addressed these concerns partially by clarifying the scope of the theory and adding empirical evidence on stability and diversity. This strengthened my confidence in the paper’s value without fully removing the underlying limitations, so I maintain my weak accept recommendation.

**Key Questions For Authors:**

1. Regarding Assumptions B.1/B.2: Can you provide empirical evidence from your molecular experiments tracking gradient variance or bias to show that the neural solver's approximation errors actually meet these strict conditions?
2. For Theorem 6.1: How sensitive is the "exact score retention" to practical discretization and truncation near $t=1$, especially since score scaling is demonstrably problematic there?
3. To address mode collapse concerns, can you provide quantitative diversity/coverage metrics for the reward-guided merged models (beyond just validity/energy) to prove prior support is preserved?

**Limitations:**

No, the authors have not adequately discussed their limitations. It does not acknowledge the gap between the idealized assumptions required for the theoretical guarantees (Theorems 6.1 and 6.2, plus Appendix B assumptions) and the messy reality of discrete-time, finite-sample, neural approximations.

**Strengths And Weaknesses:**

**Strengths:**

1. Formulating flow merging via density operators cleanly bridges distribution matching and KL-regularized control. Furthermore, Proposition 1 elegantly reduces the Union operator's multiple reverse-KL terms into a single divergence against a mixture prior, enabling efficient single-critic training.


2.  The authors identify a key practical issue: the Intersection gradient's score term diverges as $t \to 1$. Lifting the objective to the full noised flow process with time weights $\lambda_t$ is a principled, stable solution for boundary scaling.


3.  Beyond 2D toy settings, evaluating RFM on FlowMol2 (molecular design) and ETFlow (conformer generation) demonstrates the framework's practical utility in high-dimensional, structured scientific domains.



**Weaknesses:**

1. The "provable" convergence (Theorem 6.2) relies on strong stochastic approximation assumptions: precompactness of dual iterates (Assump. B.1) and Robbins-Monro conditions for solver noise/bias (Assump. B.2). These are hard to justify for deep neural solvers like Adjoint Matching, and the paper lacks empirical diagnostics to support them.


2. Theorem 6.1 claims perfect score retention, assuming the forward process perfectly converges to standard Gaussian noise. In practice, finite horizons and discretization near $t=1$ make this terminal condition only approximate, weakening the "exact" retention premise.


3.  Reward-guidance often causes mode collapse. While the authors report validity and MAE, they omit quantitative diversity or coverage metrics for the reward-guided merged models, leaving it unclear how much prior support is actually preserved.

---

> ### Author Rebuttal · Authors · 2026-03-31
>
> We thank the Reviewer for regarding our contributions as clean and elegant, and regarding our experimental evaluation as demonstrating the framework's practical utility in high-dimensional, structured scientific domains. In the following, we sharply address the concerns raised by the Reviewer referring to the new experimental results at this [url]: https://figshare.com/s/ddaeda69cd3983acd948.
>
> **Assumptions B.1/B.2**
>
> We agree that the guarantees in Sec. 6 are conditional, and in the revision we will make their scope more explicit. Our intent is not to claim that every deep implementation of Adjoint Matching automatically satisfies such assumptions, but rather that, under these standard inexact-oracle conditions, Algorithm 1 admits a rigorous asymptotic interpretation as mirror descent in probability space.
>
> - Assumption B.1 should be understood as a standard regularity condition ensuring that the dual trajectory remains in a stable regime. While this is a theoretical assumption, in practice one can partially probe its plausibility by monitoring whether the iterates remain numerically stable and do not exhibit pathological behavior throughout optimization, which we show to a certain extent for toy settings within Fig. 2.d and 2.h, and within Appendix F under "Approximate fine-tuning oracle" (Fig. 6), as well as in Fig. 7 (Apx. G.2) for molecular design. We will clarify this point in the updated manuscript.
>
> - The noise/bias assumptions in Assumption B.2 depend on the quality of the score oracle returned by the fine-tuning solver, which in practice is governed by standard design choices such as optimization budget, solver hyperparameters, and the number of Adjoint Matching (AM) iterations, quantities that can be improved at the expense of additional computational cost. In our experiments, we treat the number of AM iterations as a hyperparameter and monitor the resulting trajectories and sample quality; when the solver output has good qualitative and quantitative behavior, this provides partial evidence that the induced oracle error is sufficiently small for the optimization signal to remain nontrivial. As suggested by the Reviewer, we further evaluated in a molecular design task with FlowMol the empirical variance of the energy (used as test function) associated to fine-tuned models returned by the AM oracle conditioned on previous iterations, see [url, Fig. 3], showing that the conditional variance is well-behaved over iterates of RFM, further supporting the plausibility of such assumptions. We emphasize that this issue is not unique to our framework: any method using learned score or control oracles faces similar approximation limits. Our contribution is to show that these established fine-tuning frameworks, already empirically supported, extend to reward-guided merging. We will also clarify that these assumptions are required only for the current proof of Theorem 6.2; if they fail, the proof no longer applies, rather than implying that Problem (5) is unsolvable.
>
> **"Perfect" score retention**
>
> - We agree with the reviewer that Theorem 6.1 is an idealized exact statement. The theorem assumes that the forward process converges exactly to standard Gaussian noise, whereas in practice finite horizons and discretization imply that this terminal condition is only approximate. We will revise the wording to emphasize that Theorem 6.1 formalizes an exact oracle statement in the continuous-time idealization.
>
> - At the same time, the statement can be extended to an approximate-retention variant. Intuitively, if the terminal law of the forward process is only approximately Gaussian, then one should obtain approximate score retention, with an error term depending on the discrepancy between the actual terminal law and the Gaussian target. This is consistent with standard stability results for Schr\"odinger bridges with respect to perturbations of the endpoint marginals [1,2]. We omitted this variant in the current draft for clarity of exposition, but we will mention this extension explicitly in the revision so that the theorem is not over-interpreted as requiring literal exactness in practical finite-step implementations.
>
> **Assessment of diversity metrics**
>
> We thank the Reviewer for raising this important point. We extended the experimental evaluation to investigate mode collapse behavior by evaluating the similarities of generated molecules. We assess diversity via average Tanimoto similarity, a standard measure of molecular distance. Our results in [url, Fig. 4], demonstrate that both diversity and coverage remain stable for merged models and no significant model collapse is observed.
>
> **References**
>
> [1] Stability of Entropic Optimal Transport and Schrödinger Bridges, Espen Bernton et al., 2022.
>
> [2] Lipschitz Continuity of the Schrödinger Map in Entropic Optimal Transport, Guillaume Carlier et al., 2024.

---

> > ### Author Rebuttal · Reviewer_KLp5 · 2026-04-03
> >
> > Thank you for providing further clarification. I appreciate the authors' efforts and will incorporate this rebuttal into my final evaluation

---

### Official Review · Reviewer_tBqT · 2026-03-12

**Soundness:** 4
**Presentation:** 3
**Significance:** 2
**Originality:** 3
**Overall Recommendation:** 4
**Confidence:** 3

**Summary:**

The paper proposes a unified framework for combining pretrained generative models and reward-guided adaptation by defining density operators over probability distributions. The key idea is to view both reward-guided fine-tuning and model merging as instances of the same optimization problem: finding a distribution that balances maximizing a reward function while remaining close to a set of pretrained generative priors. Under this formulation, reward-guided fine-tuning appears as a special case with a single prior, while different divergence choices between distributions induce operators that behave like intersection, union, or interpolation of generative models. To solve this optimization problem, the authors introduce Reward-Guided Flow Merging, an iterative algorithm based on mirror descent in distribution space that repeatedly fine-tunes the model using a surrogate reward derived from the objective. The paper provides theoretical convergence guarantees under standard stochastic optimization assumptions and demonstrates the framework on toy density merging tasks and molecular design applications.

**Compliance With Llm Reviewing Policy:**

Affirmed.

**Final Justification:**

See review and rebuttal. No special concerns.

**Key Questions For Authors:**

See weaknesses.

**Limitations:**

Yes.

**Strengths And Weaknesses:**

**Strengths:**
- The formulation is flexible and allows different divergences $D(\cdot\|\cdot)$ to induce different types of operators (e.g., intersection-like, union-like, or interpolation behaviors). This provides an interesting conceptual lens for composing generative models and reasoning about their interactions.
- The proposed algorithmic approach offers an elegant way to reduce the density operator optimization problem to repeated reward-guided fine-tuning steps using the first variation $\delta G$ as a surrogate reward.
- The paper provides non-trivial theoretical analysis showing that, under assumptions on the stochastic optimization procedure, the induced sequence of terminal distributions converges almost surely to stationary points of the objective.

**Weaknesses:**
- The paper proposes a very general framework for defining density operators over generative models using arbitrary divergences and reward functionals. However, the practicality of this framework appears substantially narrower than the formulation suggests. The proposed mirror-descent algorithm requires computing the first variation of the objective functional, which in practice depends on quantities such as score functions, critic networks, or Wasserstein dual potentials depending on the chosen divergence. As a result, the framework is mathematically general but only directly implementable in settings where these quantities are tractable or can be approximated. It is not clear to me which classes of divergences and operators are practically realizable and which remain primarily conceptual.

- The algorithm assumes access to a reward-guided fine-tuning procedure that approximately solves subproblems of the form
$\max_{\pi} \mathbb{E}_{x \sim \pi}[r(x)]$ at each mirror-descent iteration, where the reward $r(x)$ is derived from the first variation $\delta G$ of the objective functional. The convergence results rely on this procedure behaving as a stochastic gradient oracle in distribution space. In practice, however, reward-guided fine-tuning of diffusion or flow models is itself a difficult optimization problem and may introduce bias or optimization error. If the solver does not accurately follow the functional gradient, the mirror-descent interpretation and the associated convergence guarantees may no longer hold. The paper does not analyze the sensitivity of the method to such solver inaccuracies. I understand that this analysis might be intractable, but it might be worth analyzing it explicitly in some ablation experiments.

- The empirical evaluation mainly demonstrates that the proposed operators behave qualitatively as intended, but it not clear whether the method provides practical advantages over simpler baselines on the downstream tasks. The experiments often illustrate that the merged distribution approximately matches the conceptual operator, rather than showing consistent improvements in task performance. Given the generality of the proposed framework, stronger empirical evidence comparing against alternative merging or reward-guided fine-tuning strategies would help clarify when the proposed approach provides a clear benefit.

- The paper interprets different divergence choices as implementing logical operators over generative models. While these interpretations are intuitive, their validity depends on assumptions about the supports and relative densities of the underlying distributions. For example, minimizing $D_{\mathrm{KL}}(p \,\|\, q)$ or $D_{\mathrm{KL}}(q \,\|\, p)$ does not in general correspond to strict set-theoretic intersection or union unless additional conditions hold. Therefore, the logical semantics of these operators appear to be heuristic rather than exact.

---

> ### Author Rebuttal · Authors · 2026-03-31
>
> We thank the Reviewer for regarding our formulation flexible and interesting, the algorithm elegant, and the analysis non-trivial. In the following, we sharply address the concerns raised by the Reviewer referring to the new experimental results at this [url]: https://figshare.com/s/ddaeda69cd3983acd948.
>
> **Practicality of the framework**
>
> We thank the Reviewer for raising this point. Indeed, there are several cases. As discussed in Sec. 4, the classic KL, which is arguably the most common divergence used in practice, allows for closed-form gradients, rendering the method extremely practical. Other common divergences (e.g., reverse KL, Wasserstein) also allow for practical f-GAN-based gradients. While the more general case is not necessarily practical, we believe that the proposed scheme is practical at least for arguably the most common divergences, as the aforementioned ones, which allow to implement density operators of high practical relevance, including intersection, union, and interpolation, as discussed in Sec. 3.
>
> **Analysis for Approximate Stochastic Gradient**
>
> We believe this question is important. Nonetheless, the Reviewer might not have found certain parts of the paper already discussing these aspects, which we will make sure to properly refer in the main paper within the updated manuscript.
>
> 1. Our theoretical analysis does account for both noise and bias in the reward-guided fine-tuning oracle. In fact, as stated in Sec. 6, we account for "approximate mirror iterates", not exact ones. This is a fundamental difference from standard mirror descent analysis [e.g., 1], which does not account for approximate iterates. As mentioned within Sec. 6, a similar analysis has been previously employed for the Sinkhorn algorithm in computational optimal transport [2], whose iterates are also approximate.
>
> 2. The submitted work contains an experimental sensitivity analysis of the reward-guided fine-tuning oracle within Appendix F "Computational Complexity, Cost, and Approximate Fine-Tuning Oracles". In particular, it shows that RFM can work reliably with an approximate oracle, and analyses how the degree of oracle approximation can be traded-off with the number of RFM iterations $K$.
>
> **Model Merging Baselines**
>
> To clarify the positioning against alternatives flow merging baselines, we compared RFM against Diffusion Soup [1], which is based on the common weight averaging scheme [2].
>
> 1. Weight averaging methods [e.g., 2] require all prior models to share the same architecture. By contrast, RFM does not impose this restriction. To assess this capability of fundamental practical importance, we ran new experiments using RFM to merge FlowMol and SemlaFlow [4] on GEOM-Drugs computing their intersection. RFM successfully merges them despite FlowMol and SemlaFlow having different architectures, see [url, Fig. 1].
>
> 2. Weight averaging [e.g., 2] does not allow interpretable control over the merged density. RFM explicitly controls whether the merged model corresponds to the intersection (Fig. 2.b) or the union (Fig. 2.f for the models in Fig. 2.e) etc.. We ran Diffusion Soup [2] on this illustrative problem: it produces a broken model, as shown in [url, Fig. 2]. By contrast, our framework yields predictable, controllable merging operations, whereas weight averaging leads to an exploded density even in this simple 2D setting.
>
> 3. Prior weight averaging methods do not enable task-aware merging, whereas our framework does through reward-guided merging.
>
> We believe these additional empirical and conceptual analyses clarify the positioning of RFM relative to widely adopted current flow merging methods.
>
> **Interpretation of logical operators**
>
> As mentioned in Sec. 3, we use intersection (analogously, union etc.) in the common product-of-experts sense used in some recent ML works [e.g., 3,4,5], where distributions are multiplied and renormalized to emphasize regions jointly supported by all experts. We agree with the Reviewer that the set-logic interpretation of the term is not guaranteed, and depends on assumptions on the models' densities supports. We thank the Reviewer for raising this point regarding clarity, and will clarify it within the updated manuscript.
>
> **References**
>
> [1] Relatively-Smooth Convex Optimization by First-Order Methods, and Applications, Haihao Lu, SIAM Journal on Optimization, 2017.
>
> [2] Sinkhorn flow as mirror flow: A continuous-time framework for generalizing the sinkhorn algorithm, Mohammad Reza Karimi, AISTATS, 2024.
>
> [3] Sampling-based constrained motion planning with products of experts, Amirreza Razmjoo et al., The International Journal of Robotics Research, 2026.
>
> [4] CMAD: Cooperative Multi-Agent Diffusion via Stochastic Optimal Control, Riccardo Barbano et al., 2026.
>
> [5] Unsupervised Decomposition and Recombination with Discriminator-Driven Diffusion Models, Archer Wang et al., 2026.

---

> > ### Author Rebuttal · Reviewer_tBqT · 2026-04-03
> >
> > Many thanks to the authors for a rebuttal that clarifies that the framework is practical for common divergences (e.g., KL-based objectives). It is helpful, although the broader generality still remains somewhat conceptual. The discussion of approximate mirror descent is also appreciated, though the practical behavior of the fine-tuning oracle remains an important factor. The additional experiments help illustrate the flexibility of the approach, but I still find that the empirical evidence focuses more on demonstrating operator behaviour than on establishing clear advantages over strong baselines on downstream tasks.
> >
> > I still think that the paper should be accepted so I will keep my score of 4.

---

> > > ### Author Response · Authors · 2026-04-06
> > >
> > > We thank the Reviewer for the thoughtful follow-up and for clarifying their assessment. Regarding the point on baselines, in the revised manuscript we will make the comparison to weight-averaging/Diffusion-Soup more explicit, including the ability to merge models with different architectures, so that the main practical distinction is clearer.

---

### Official Review · Reviewer_8i3F · 2026-03-23

**Soundness:** 3
**Presentation:** 3
**Significance:** 3
**Originality:** 3
**Overall Recommendation:** 4
**Confidence:** 2

**Summary:**

The paper unifies two previously separate problems, reward-guided fine-tuning and flow model merging, under a single probability-space optimization framework. The key formulation expresses both as optimizing a functional that balances expected reward against weighted divergences to multiple pre-trained models. By varying divergence type, this recovers intersection (AND), union (OR), and interpolation operators over generative model densities, as well as reward-guided variants and complex compositions.To solve this otherwise intractable distributional objective, the authors propose Reward-Guided Flow Merging (RFM), a mirror descent scheme that reduces each iteration to a standard fine-tuning step via surrogate rewards derived from the objective's first variation. They also prove that SOC-based fine-tuning retains score information, enabling a rigorous convergence analysis that yields first-of-their-kind guarantees for flow merging. Experiments on synthetic settings, molecular design, and conformer generation validate the approach.

**Compliance With Llm Reviewing Policy:**

Affirmed.

**Final Justification:**

My concerns have been addressed by the rebuttal. I maintain my positive assessment in favour of acceptance.

**Key Questions For Authors:**

- Conformer generation is evaluated on a single molecule. Can the authors demonstrate consistent results across a broader set of molecular systems to support the generalization claim?
- The merging objective is enforced only at the distributional level. Can the authors report per-sample constraint satisfaction rates and discuss how severe the gap between distributional and per-sample feasibility is in the molecular experiments?
- The placeholder citations make it difficult to distinguish this paper's contributions from those of CFO, CVaR, and FE. Can the authors clarify which technical contributions, particularly around the convergence analysis and first-variation gradient derivations, belong exclusively to this work?
- No baselines beyond the pre-trained models are included. Can the authors compare RFM against simpler alternatives such as score interpolation or weight-space averaging, even in a 2D toy setting, to justify the added complexity of RFM?

**Limitations:**

See weaknesses.

**Strengths And Weaknesses:**

**Strengths**

- The probability-space framework cleanly unifies reward-guided fine-tuning and flow merging as limit cases, providing a principled basis for AND/OR/interpolation operators previously absent in the literature.
- The 2D illustrative experiments provide visually interpretable evidence that RFM correctly implements each operator, complementing the higher-dimensional molecular results.
- Theorem 6.1 rigorously justifies the score retention assumption that prior control-based analyses implicitly relied on, making it a useful standalone contribution.
- The K/N trade-off analysis honestly characterizes RFM's computational behavior and gives practitioners actionable guidance.

**Weaknesses**

- The empirical evaluation is narrow. Conformer generation is assessed on a single molecule, and molecular design is limited to one model (FlowMol) and one dataset (GEOM-Drugs), making it difficult to assess generalizability across different chemical spaces or model architectures.
- The gap between distributional constraint enforcement and per-sample validity is not discussed. A non-trivial validity drop is observed across models in Table 1 (e.g., 76% to 68%), yet the paper provides no analysis of how this should be interpreted or mitigated.
- No merging baselines are included beyond the pre-trained models themselves, leaving open whether simpler alternatives such as weight-space averaging or naive score mixing would achieve comparable results at lower cost.
- The boundary with concurrent submissions from what appears to be the same group is unclear. Key technical elements including mirror descent over flow processes and score retention are shared with CFO and FE, making it difficult to assess which contributions are unique to this work.

---

> ### Author Rebuttal · Authors · 2026-03-31
>
> We thank the Reviewer for regarding our framework principled, absent in the literature, the visual results useful, and Theorem 6.1 rigorous. In the following, we address the concerns raised by the Reviewer referring to the new experimental results at this [url]: https://figshare.com/s/ddaeda69cd3983acd948.
>
> **Narrow experimental evaluation**
>
> The version submitted contained molecular design tasks over two datasets, namely QM9 and GEOM-Drugs. Nonetheless, we have substantially extended the empirical evaluation of molecular design and conformer generation as suggested. We performed new experiments with an additional model, SemlaFlow [3] showing that RFM can merge models with different architectures (FlowMol and SemlaFlow, see [url, Fig. 1]). We also strengthened the analysis by adding further diversity metrics [url, Fig. 4], providing evidence that RFM does not suffer from mode collapse. As suggested, we have also extended conformer generator merging beyond a single molecular system with successful results [url, Fig. 5 and Table 1]. These results will be included within the updated manuscript.
>
> **Validity measure**
>
> 1. (Distributional constraint) In the current work, we evaluate expected sample validity. We are not sure what the Reviewer refers to by "distributional constraint". If the Reviewer meant a different validity notion, we would appreciate clarification.
>
> 2. (Validity drop) We believe the Reviewer misunderstood the validity estimates reported in Table 1:  $68\%$ is the validity of one of the two pre-trained models (reported as PRE-2). All final merged models via RFM achieve high validity ($72.78$ - $74.34$).
>
> **Model Merging Baselines**
>
> To clarify the positioning against alternatives flow merging baselines, we compared RFM against Diffusion Soup [1], which is based on weight averaging [2] as suggested.
>
> 1. Weight averaging methods [e.g., 2] require all prior models to share the same architecture. By contrast, RFM does not impose this restriction. To assess this capability of fundamental practical importance, we ran new experiments using RFM to merge FlowMol and SemlaFlow [4] on GEOM-Drugs computing their intersection. RFM successfully merges them despite FlowMol and SemlaFlow having different architectures, see [url, Fig. 1].
>
> 2. Weight averaging [e.g., 2] does not allow interpretable control over the merged density. RFM explicitly controls whether the merged model corresponds to the intersection (Fig. 2.b) or the union (Fig. 2.f for the models in Fig. 2.e) etc.. We ran Diffusion Soup [2] on this illustrative problem: it produces a broken model, as shown in [url, Fig. 2]. By contrast, our framework yields predictable, controllable merging operations, whereas weight averaging leads to an exploded density even in this simple 2D setting.
>
> 3. Prior weight averaging methods do not enable task-aware merging, whereas our framework does through reward-guided merging.
>
> We believe these additional empirical and conceptual analyses clarify the positioning of RFM relative to widely adopted current flow merging methods.
>
> **Concurrent and prior works**
> We would like to clarify points and their relation to concurrent and prior work:
>
> 1. CFO studies constrained generative optimization and uses neither mirror flow nor score retention. Although both works concern post-training of flow processes, they differ substantially in goal, formulation, and method. To our knowledge, there is no relevant technical or conceptual connection.
>
> 2. CVaR studies tail-aware flow adaptation. Like RFM, it addresses distributional fine-tuning, but with a different goal and different technical tools in formulation, algorithm, and analysis: CVaR targets risk-averse fine-tuning, not model merging. Beyond the broader theme of distributional post-training of flow processes, already discussed within the related works, we are not aware of a relevant technical connection.
>
> 3. FE uses mirror descent for an entropic exploration objective, whereas here we use it for flow merging. Mirror descent, like gradient descent, is a general optimization scheme applicable to many objectives. Importantly, FE belongs to the class of prior works discussed in Sec. 6 that implicitly assume score retention and would therefore benefit from new theoretical results on this point. In particular, Theorem 6.1, a central contribution of this work, fundamentally strengthens prior convergence analyses, including the one within FE, and is, to our knowledge, novel relative to both prior and concurrent work.
>
> **References**
>
> [1] Diffusion Soup: Model Merging for Text-to-Image Diffusion Models, Benjamin Biggs et al., 2024.
>
> [2] Model soups: averaging weights of multiple fine-tuned models improves accuracy without increasing inference time, Mitchell Wortsman et al., 2022.
>
> [3] SemlaFlow -- Efficient 3D Molecular Generation with Latent Attention and Equivariant Flow Matching, Ross Irwin et al., 2025.
>
> [4] RDKit: Open-source cheminformatics, Greg Landrum et al., 2025.

---

> > ### Author Rebuttal · Reviewer_8i3F · 2026-04-03
> >
> > I thank the authors for the rebuttal. My concerns have been addressed by the rebuttal. I maintain my positive assessment in favour of acceptance.

---

### Official Review · Reviewer_RAaW · 2026-03-24

**Soundness:** 3
**Presentation:** 3
**Significance:** 2
**Originality:** 3
**Overall Recommendation:** 5
**Confidence:** 3

**Summary:**

the authors propose a framework to perform flow merging and finetuning of flow models at the same time called RFM. They demonstrate their framework via a diversity of operators like AND/OR/…, provide theoretical guarantees for some of it and demonstrate its performance on small molecule datasets.

**Compliance With Llm Reviewing Policy:**

Affirmed.

**Final Justification:**

The authors addressed my main concerns, which is why I raise my score to accept.

**Key Questions For Authors:**

[Q1] Adjoint Matching and other SOC methods often tend to be sensitive to hyperparameters and other small variances; how complex was tuning of the method for the demonstrated tasks/how easy is it to apply this technique to a new very different flow model?

**Limitations:**

yes

**Strengths And Weaknesses:**

[S1] elegant unification with valid theoretical underpinnings: the authors provide a nice joint framework for the separate tasks of flow merging and model finetuning, and demonstrate its perofrmance via theoretical guarantees as well as applications.

[W1] The applications are all rather toy-scale, it would be great to see approaches for more relevant models/systems to demonstrate broad applicability/scalability of the method, also strengthening the efficiency claim.

[W2] Benchmarks against inference-time composition approaches: in papers like Ito Superposition [1] a similar effect to flow merging is achieved at inference time, giving a lot more flexibility. The authors should ideally compare their finetuning approach directly against these approaches to show that their finetuning outperforms these simpler inference time methods.

References

[1] Skreta, Marta, et al. "The Superposition of Diffusion Models Using the It\^ o Density Estimator." arXiv preprint arXiv:2412.17762 (2024).

---

> ### Author Rebuttal · Authors · 2026-03-31
>
> We thank the Reviewer for regarding our framework as elegant, and with valid theoretical underpinnings. In the following, we sharply address the concerns raised by the Reviewer also referring to the new experimental results at this [url]: https://figshare.com/s/ddaeda69cd3983acd948.
>
> **Toy-scale applications and diversity of datasets and models**
>
> We believe the presented experimental results on real-world data are not toy-scale as they show that the method scales to recent state-of-the-art models, FlowMol and ETFlow, on both QM9 and the drug-like GEOM-Drugs dataset, which is typically regarded as challenging. Thus, we believe our evaluation already shows promising performance on relevant models and standard benchmarks in ML for molecular design. Nonetheless, we followed the Reviewer’s suggestion, and ran new experiments evaluating RFM for molecular design with a third model, SemlaFlow [4], on GEOM-Drugs; these results are discussed below. We also added diversity metrics, as suggested by Reviewer KLp5, indicating that RFM does not appear to suffer from mode collapse (more details are provided in the response to Reviewer KLp5), and extended the conformer generation task to multiple systems [url, Fig. 5 and Table 1]. While we agree that applying the method to a high-impact real-world task, e.g., in biochemical design, is a natural next step, this lies beyond the scope of the present work, which is primarily of mathematical and algorithmic nature.
>
> **Baselines**
>
> We thank the Reviewer for requesting clearer positioning against alternatives. Nonetheless, our work studies model merging, not inference-time composition, and the two address fundamentally different problems. Merging combines multiple models into one, allowing deletion of prior models, whereas inference-time composition retains them and only combines sampling processes. This distinction matters because deleting prior models enables continual model combination, removes per-sample sampling costs, and thus renders merging inherently needed and superior for several real-world applications [2]. We therefore believe comparison to inference-time composition would be misleading. Following Reviewer 8i3F’s suggestion, we compared against Diffusion Soup [2], based on the widely adopted weight-averaging scheme  [2,3] for flow merging, and will include the results and discussion in the revised manuscript. In summary:
>
> 1. Weight averaging methods [e.g., 2] require all prior models to share the same architecture. By contrast, RFM does not impose this restriction. To assess this capability of fundamental practical importance, we ran new experiments using RFM to merge FlowMol and SemlaFlow [4] on GEOM-Drugs to compute their intersection (i.e., the AND operator). RFM successfully merges them despite FlowMol and SemlaFlowad having different architectures, see [url, Fig. 1].
>
> 2. Weight averaging schemes [e.g., 2] do not allow interpretable control over the merged density. RFM explicitly controls whether the merged model corresponds to the intersection (Fig. 2.b) or the union (Fig. 2.f for the models in Fig. 2.e) etc., whereas weight averaging cannot. We ran Diffusion Soup [2] on this illustrative problem; it produces a broken model, as shown in [url, Fig. 2] By contrast, our framework yields predictable, controllable merging operations, whereas weight averaging leads to a non-sensical density even in this simple 2D setting.
>
> 3. Prior weight averaging methods do not enable task-aware merging, whereas our framework does through reward-guided merging.
>
> We believe these additional empirical and conceptual analyses clarify the positioning of RFM relative to prior flow merging methods.
>
> **Q1: complexity of tuning and adoption**
>
> Since we show that (reward-guided) model merging can be reduced to sequential fine-tuning with any off-the-shelf gradient-based fine-tuning method, RFM inherits its tuning complexity from the oracle it uses and is directly applicable. While reward-guided adaptation remains challenging, as it is a high-dimensional optimal control problem, recent work has made significant progress toward scalable approaches. Improving these solvers is orthogonal to our contribution, and future advances in reward-guided fine-tuning will be immediately usable within RFM.
>
> **References**
>
> [1] The Superposition of Diffusion Models Using the Ito Density Estimator, Marta Skreta et al., ICLR 2024.
>
> [2] Diffusion Soup: Model Merging for Text-to-Image Diffusion Models, Benjamin Biggs et al., 2024.
>
> [3] Model soups: averaging weights of multiple fine-tuned models improves accuracy without increasing inference time, Mitchell Wortsman et al., 2022.
>
> [4] SemlaFlow -- Efficient 3D Molecular Generation with Latent Attention and Equivariant Flow Matching, Ross Irwin et al., 2025.

---

> > ### Author Rebuttal · Reviewer_RAaW · 2026-04-04
> >
> > Thanks for the replies, I will adjust my score accordingly.

---

### Decision · Program_Chairs · 2026-04-30

**Decision:**

Accept (regular)

**Comment:**

This paper presents a unified probabilistic framework that brings together reward-guided fine-tuning and model merging for continuous-time generative models by introducing density operators.  This paper makes a solid contribution to the generative modeling community.